# LEARNING EXPLAINABLE MODELS USING ATTRIBUTION PRIORS

## ABSTRACT

Two important topics in deep learning both involve incorporating humans into the modeling process: *Model priors* transfer information from humans to a model by regularizing the model's parameters; *Model attributions* transfer information from a model to humans by explaining the model's behavior. Previous work has taken important steps to connect these topics through gradient regularization, but we find that these methods do not successfully align a model's behavior with human intuition. We develop an efficient and theoretically grounded feature attribution method, *expected gradients*, and a novel framework, *attribution priors*, to enforce prior expectations about a model's behavior during training. We demonstrate that attribution priors are broadly applicable in three different domains: image data, gene expression data, and health care data. Our experiments show that models trained with attribution priors are more intuitive and achieve better generalization performance than both equivalent baselines and existing methods to regularize model behavior.

## 1 INTRODUCTION

Recent work on interpreting machine learning models has focused on *feature attribution methods*. Given an input feature, a model, and a prediction on a particular sample, such methods assign a number to the input feature that represents how important the input feature was for making the prediction. Previous literature about such methods has focused on the axioms they should satisfy (Lundberg and Lee, 2017; Sundararajan et al., 2017; Štrumbelj and Kononenko, 2014; Datta et al., 2016), and how attribution methods can give us insight into model behavior (Lundberg et al., 2018a;b; Sayres et al., 2019; Zech et al., 2018). These methods can be an effective way of revealing problems in a model or a dataset. For example, a model may place too much importance on undesirable features, rely on many features when sparsity is desired, or be sensitive to high frequency noise. In such cases, we often have a prior belief about how a model should treat input features, but for neural networks it can be difficult to mathematically encode this prior in terms of the original model parameters.

Ross et al. (2017b) introduce the idea of regularizing explanations to train models that better agree with domain knowledge. Given a binary variable indicating whether each feature should or should not be important for predicting on each sample in the dataset, their method penalizes the gradients of unimportant features. However, two drawbacks limit the method's applicability to real-world problems. First, gradients don't satisfy the theoretical guarantees that modern feature attribution methods do (Sundararajan et al., 2017). Second, it is often difficult to specify which features should be important in a binary manner. More recent work has stressed that incorporating intuitive, *human* priors will be necessary for developing robust and interpretable models (Ilyas et al., 2019). Still, it remains challenging to encode meaningful, human priors like "have smoother attribution maps" or "treat this group of features similarly" by penalizing the gradients or parameters of a model.

In this work, we propose an expanded framework for encoding abstract priors, called *attribution priors*, in which we directly regularize differentiable functions of a model's axiomatic feature attributions during training. This framework, which can be seen as a generalization of gradient-based regularization (LeCun et al., 2010; Ross et al., 2017b; Yu et al., 2018; Jakubovitz and Giryes, 2018; Roth et al., 2018), can be used to encode meaningful domain knowledge more effectively than existing methods. Furthermore, we introduce a novel feature attribution method - *expected gradients* - which extends integrated gradients (Sundararajan et al., 2017), is naturally suited to being regularized

under an attribution prior, and avoids hyperparameter choices required by previous methods. Using attribution priors, we build improved deep models for three different prediction tasks. On images, we use our framework to train a deep model that is more interpretable and generalizes better to noisy data by encouraging the model to have piecewise smooth attribution maps over pixels. On gene expression data, we show how to both reduce prediction error and better capture biological signal by encouraging similarity among gene expression features using a graph prior. Finally, on a patient mortality prediction task, we develop a sparser model and improve performance when learning from limited training data by encouraging a skewed distribution of the feature attributions.

## 2 ATTRIBUTION PRIORS

In this section, we formally define an attribution prior, and give three example priors for different data types. Let $X \in \mathbb{R}^{n \times p}$ denote a dataset with labels $y \in \mathbb{R}^o$, where $n$ is the number of samples, $p$ is the number of features, and $o$ is the number of outputs. In standard deep learning we aim to find optimal parameters $\theta$ by minimizing loss, subject to a regularization term $\Omega'(\theta)$ on the parameters:

$$\theta = \mathrm{argmin}_\theta \mathcal{L}(\theta; X, y) + \lambda' \Omega'(\theta).$$

For some model parameters $\theta$, let $\Phi(\theta, X)$ be a feature attribution method, which is a function of $\theta$ and the data $X$. Let $\phi_i^\ell$ be the feature importance of feature $i$ in sample $\ell$. We formally define an *attribution prior* as a scalar-valued penalty function of the feature attributions, $\Omega(\Phi(\theta, X))$, which represents a log-transformed prior probability distribution over possible attributions.

$$\theta = \mathrm{argmin}_\theta \mathcal{L}(\theta; X, y) + \lambda \Omega(\Phi(\theta, X)),$$

where $\lambda$ is the regularization strength. We note that the attribution prior function $\Omega$ is agnostic to the attribution method $\Phi$. While in Section 3 we propose a feature attribution method for attribution priors, other attribution methods can be used. This includes existing methods like integrated gradients or simply the gradients themselves. In the latter case, we can see the method proposed in Ross et al. (2017b) as a specific instance of an attribution prior:

$$\theta = \mathrm{argmin}_\theta \mathcal{L}(\theta; X, y) + \lambda'' ||A \odot \frac{\partial \mathcal{L}}{\partial X}||_F^2$$

where the attribution method $\Phi(\theta, X)$ is the gradients of the model, represented by the matrix $\frac{\partial \mathcal{L}}{\partial X}$ whose $\ell, i$th entry is the gradient of the loss at the $\ell$th sample with respect to the $i$th feature. $A$ is a binary matrix indicating which features should be penalized in which samples.

Often, however, we do not know which features are important in advance. Instead, we can define different attribution priors for different tasks depending on the data and our domain knowledge. To demonstrate how attribution priors can capture human intuition in a variety of domains, in the following sections we first define and then apply three different priors for three different data types.

### 2.1 PIXEL ATTRIBUTION PRIOR FOR IMAGE CLASSIFICATION

Prior work on interpreting image models has focused on creating pixel attribution maps, which assign a value to each pixel indicating how important that pixel was for a model's prediction (Selvaraju et al., 2017; Sundararajan et al., 2017). These attribution maps can be noisy and often highlight seemingly unimportant pixels in the background. Such attributions can be difficult to understand, and may indicate the model is vulnerable to adversarial attacks (Ross and Doshi-Velez, 2018). Although we may desire a model with smoother attributions, existing methods only post-process attribution maps and do not change model behavior (Smilkov et al., 2017; Selvaraju et al., 2017; Fong and Vedaldi, 2017). Such techniques may not be faithful to the original model (Ilyas et al., 2019). In this section, we describe how to apply our framework to train image models with naturally smoother attributions.

To regularize pixel-level attributions, we use the following intuition: neighboring pixels should have a similar impact on an image model's output. To encode this intuition, we apply a total variation loss on pixel-level attributions as follows:

$$\Omega_{\text{pixel}}(\Phi(\theta, X)) = \sum_\ell \sum_{i,j} |\phi_{i+1,j}^\ell - \phi_{i,j}^\ell| + |\phi_{i,j+1}^\ell - \phi_{i,j}^\ell|,$$

where $\phi_{i,j}^{\ell}$ is the attribution for the $i, j$-th pixel in the $\ell$-th training image. Including the $\lambda$ scale factor, this penalty is equivalent to placing a Laplace$(0, \lambda^{-1})$ prior on the differences between adjacent pixel attributions. For further details, see Bardsley (2012) and the Appendix.

## 2.2 GRAPH ATTRIBUTION PRIOR FOR GENE EXPRESSION DATA

In the image domain, our attribution prior took the form of a penalty encouraging smoothness over adjacent pixels. In other domains, we may have prior information about specific relationships between features that can be encoded as an arbitrary graph (such as social networks, knowledge graphs, or protein-protein interactions). For example, prior work in bioinformatics has shown that protein-protein interaction networks contain valuable information that can be used to improve performance on biological prediction tasks (Cheng et al., 2014). These networks can be represented as a weighted, undirected graph. Formally, say we have a weighted adjacency matrix $W \in \mathbb{R}_{+}^{p \times p}$ for an undirected graph, where the entries encode our prior belief about the pairwise similarity of the importances between two features. For a biological network, $W_{i,j}$ encodes either the probability or strength of interaction between the $i$-th and $j$-th genes (or proteins). We can encourage similarity along graph edges by penalizing the squared Euclidean distance between each pair of feature attributions in proportion to how similar we believe them to be. Using the graph Laplacian ($L_G = D - W$), where $D$ is the diagonal degree matrix of the weighted graph this becomes:

$$\Omega_{\text{graph}}(\Phi(\theta, X)) = \sum_{i,j} W_{i,j}(\bar{\phi}_i - \bar{\phi}_j)^2 = \bar{\phi}^T L_G \bar{\phi}.$$

In this case, we choose to penalize *global* rather than local feature attributions. So we define $\bar{\phi}_i$ to be the importance of feature $i$ across all samples in our data set, where this global attribution is calculated as the average magnitude of the feature attribution across all samples: $\bar{\phi}_i = \frac{1}{n} \sum_{\ell=1}^{n} |\phi_i^{\ell}|$. Overall, $\Omega_{\text{graph}}$ is equivalent to placing a Normal$(0, \lambda^{-1})$ prior on the differences between attributions for features that are adjacent in the graph. See Bardsley (2012) and the Appendix for details.

## 2.3 SPARSITY ATTRIBUTION PRIOR FOR FEATURE SELECTION

*Feature selection* and *sparsity* are popular ways to alleviate the curse of dimensionality, facilitate interpretability, and improve generalization by building models that use a small number of input features. A straightforward way to build a sparse deep model is to apply an L1 penalty to the first layer (and possibly subsequent layers) of the network. Similarly, the sparse group lasso (SGL) penalizes all weights connected to a given feature (Feng and Simon, 2017; Scardapane et al., 2017), while Ross et al. (2017a) penalize the gradients of each feature in the model.

These approaches suffer from two problems: First, a feature with small gradients or first-layer weights may still strongly affect the model's output (Shrikumar et al., 2017). A feature whose attribution value (e.g., integrated or expected gradient) is zero, is much less likely to have any effect on predictions. Second, successfully minimizing the L1 or SGL penalty is not necessarily the best way to create a sparse model. A model that puts weight $w$ on 1 feature is penalized more than one that puts weight $\frac{w}{2p}$ on each of $p$ features. Prior work on sparse linear regression has shown that the Gini coefficient $G$ of the weights, proportional to 0.5 minus the area under the CDF of sorted values, avoids such problems and corresponds more directly to a sparse model (Hurley and Rickard, 2009; Zonoobi et al., 2011). We extend this analysis to deep models by noting that the Gini coefficient can be written differentiably and using it to develop an attribution penalty based on the global feature attributions $\bar{\phi}_i$:

$$\Omega_{\text{sparse}}(\Phi(\theta, X)) = -2G(\Phi) = -\frac{\sum_{i=1}^{p} \sum_{j=1}^{p} |\bar{\phi}_i - \bar{\phi}_j|}{n \sum_{i=1}^{p} \bar{\phi}_i}.$$

This is similar to the total variation penalty $\Omega_{\text{image}}$, but normalized and with a flipped sign to *encourage* differences. The corresponding attribution prior is maximized when global attributions are zero for all but one feature, and minimized when attributions are uniform across features.

## 3 Expected Gradients

Here we propose a feature attribution method called *expected gradients* and describe why it is a natural choice for attribution priors. Expected gradients is an extension of integrated gradients (Sundararajan et al., 2017) with fewer hyperparameter choices. Like several other attribution methods, integrated gradients aims to explain the difference between a model's current prediction and the prediction that the model would make when given a baseline input. This baseline input is meant to represent some uninformative reference input, which represents not knowing the value of the input features. Although choosing such an input is necessary for several feature attribution methods (Sundararajan et al., 2017; Shrikumar et al., 2017; Binder et al., 2016), the choice is often made arbitrarily. For example, in image tasks, the image of all zeros is often chosen as a baseline, but doing so implies that black pixels will not be highlighted as important by existing feature attribution methods. In many domains, it is not clear how to choose a baseline that correctly represents a lack of information.

Our method avoids an arbitrary choice of baseline by modeling not knowing the value of a feature by integrating over a dataset. For a model $f$, the *integrated gradients* value for feature $i$ is defined as:

$$\text{IntegratedGradients}_i(x, x') := (x_i - x_i') \times \int_{\alpha=0}^{1} \frac{\delta f(x' + \alpha \times (x - x'))}{\delta x_i} \delta\alpha,$$

where $x$ is the target input and $x'$ is baseline input. To avoid specifying $x'$, we define the *expected gradients* value for feature $i$ as:

$$\text{ExpectedGradients}_i(x) := \int_{x'} \left( (x_i - x_i') \int_{\alpha=0}^{1} \frac{\delta f(x' + \alpha \times (x - x'))}{\delta x_i} \delta\alpha \right) p_D(x') \delta x',$$

where $D$ is the underlying data distribution. Since expected gradients is also a diagonal path method, it satisfies the same axioms as integrated gradients (Friedman, 2004). Directly integrating over the training distribution is intractable; so we instead reformulate the integrals as expectations:

$$\text{ExpectedGradients}_i(x) := \mathop{\mathbb{E}}_{x' \sim D, \alpha \sim U(0,1)} \left[ (x_i - x_i') \frac{\delta f(x' + \alpha \times (x - x'))}{\delta x_i} \right].$$

This expectation-based formulation lends itself to a natural sampling based approximation method: draw samples of $x'$ from the training dataset and $\alpha$ from $U(0, 1)$, compute the value inside the expectation for each sample, and average over samples.

**Training with expected gradients:** If we let the attribution function $\Phi$ in our attribution prior $\Omega(\Phi(\theta, X))$ be expected gradients, a good approximation during training appears to require computing an expensive Monte Carlo estimate with hundreds of extra gradient calls every training step. Ordinarily, this would make training with such attributions intractable. However, most deep learning models today are trained using some variant of batch gradient descent, in which the gradient of a loss function is approximated over many training steps using mini-batches of data. We can use a batch training procedure to approximate expected gradients over the training procedure as well. During training, we let $k$ be the number of samples we draw to compute expected gradients for each mini-batch of data. Remarkably, we find that as small as $k = 1$ suffices to regularize the explanations because of the averaging effect of the expectation formulation over many training samples. This choice of $k$ leads to every sample in the training set being used as a reference over the course of an epoch with only one additional gradient call per training step. This results in far more reference samples than the 100-200 we found necessary for reliable individual attributions (see Appendix).

## 4 Experiments

We first evaluate expected gradients by comparing it with other feature attribution methods on 18 benchmarks introduced in Lundberg et al. (2019) (Table 1). These benchmark metrics aim to evaluate how well each attribution method finds the most important features for a given dataset and model. For all metrics, a larger number corresponds to a better feature attribution method. Expected gradients significantly outperforms the next best feature attribution method ($p = 7.2 \times 10^{-5}$, one-tailed Binomial test). We provide more details and also additional benchmarks in the Appendix.

Table 1: Benchmark results on synthetic data with correlated features. Larger numbers are better for all metrics. For metric names (K = Keep, R = Remove), (P = Positive, N = Negative, A = Absolute), (M = Mean masking, R = Resample masking, and I = Impute masking) (see Appendix for details).

| Method | KPM | KPR | KPI | KNM | KNR | KNI | KAM | KAR | KAI |
|---|---|---|---|---|---|---|---|---|---|
| Expected Grad. | **3.731** | **3.800** | **3.973** | **3.615** | **3.551** | **3.873** | **0.906** | **0.903** | 0.919 |
| Integrated Grad. | 3.667 | 3.736 | 3.920 | 3.543 | 3.476 | 3.808 | 0.905 | 0.899 | **0.920** |
| Gradients | 0.096 | 0.122 | 0.099 | 0.076 | -0.112 | 0.052 | 0.838 | 0.823 | 0.887 |
| Random | 0.033 | 0.106 | 0.077 | -0.012 | -0.093 | -0.053 | 0.593 | 0.583 | 0.715 |

| Method | RPM | RPR | RPI | RNM | RNR | RNI | RAM | RAR | RAI |
|---|---|---|---|---|---|---|---|---|---|
| Expected Grad. | **3.612** | **3.575** | **3.525** | **3.759** | **3.830** | **3.683** | **0.897** | **0.885** | **0.880** |
| Integrated Grad. | 3.539 | 3.503 | 3.365 | 3.687 | 3.754 | 3.543 | 0.872 | 0.859 | 0.822 |
| Gradients | 0.035 | -0.098 | -0.020 | 0.110 | 0.105 | 0.108 | 0.729 | 0.712 | 0.616 |
| Random | -0.053 | -0.100 | -0.106 | 0.034 | 0.092 | 0.111 | 0.400 | 0.400 | 0.275 |

## 4.1 A PIXEL ATTRIBUTION PRIOR IMPROVES ROBUSTNESS TO IMAGE NOISE

We apply our $\Omega_{\text{pixel}}$ attribution prior to the CIFAR-10 dataset (Krizhevsky et al., 2009). We train a VGG16 network from scratch (Simonyan and Zisserman, 2014), and optimize hyperparameters for the baseline model without an attribution prior. To choose $\lambda$, we search over values in $[10^{-20}, 10^{-1}]$, and choose the $\lambda$ that minimizes the attribution prior penalty and achieves a test accuracy within $10\%$ of the baseline model. Figure 1 displays expected gradients attribution maps for both the baseline and the model regularized with an attribution prior on 5 randomly selected test images. In all examples, the attribution prior results in a model with visually smoother attributions. Remarkably, smoother attributions also often better highlight the structure of the target object in the image in many instances.

Recent work in understanding image classifiers has suggested that they are brittle to small domain shifts: small changes in the underlying distribution of the training and test set can result in significant drops in test accuracy (Recht et al., 2019). To simulate a domain shift, we apply Gaussian noise to images in the test set and re-evaluate the performance of the regularized model and the baseline model. As an adaptation of Ross et al. (2017b), we also compare to regularizing the total variation of gradients with the same criteria for choosing $\lambda$. For each method, we train 5 models with different random initializations. In Figure 1, we plot the mean and standard deviation of test accuracy as a function of standard deviation of added Gaussian noise. The figure shows that our regularized model is more robust to noise than both the baseline and the gradient-based model.

Although our method provides both robustness and more intuitive saliency maps, this comes at the cost of reduced test set accuracy ($0.93 \pm 0.002$ for the baseline vs. $0.85 \pm 0.003$ for pixel attribution prior model). The trade-off between robustness and accuracy that we observe is in line with previous work that suggests image classifiers trained solely to maximize test accuracy rely on features that are brittle and difficult to interpret (Ilyas et al., 2019; Tsipras et al., 2018; Zhang et al., 2019). Despite this trade-off, we find that at a stricter hyperparameter cutoff for $\lambda$ - within $1\%$ test accuracy of the baseline, rather than $10\%$ - our methods are still able to achieve modest but significant robustness relative to the baseline. For results at different hyperparameter thresholds, as well as more details on our training procedure and additional experiments on MNIST, see the Appendix.

## 4.2 A GRAPH ATTRIBUTION PRIOR IMPROVES ANTI-CANCER DRUG RESPONSE PREDICTION

Incorporating the $\Omega_{\text{graph}}$ attribution prior not only leads to a model with more reasonable attributions, but also improves predictive performance by allowing us to incorporate prior biological knowledge into the training process. We downloaded publicly available gene expression and drug response data for patients with acute myeloid leukemia (AML, a type of blood cancer) and tried to predict patients' drug response from their gene expression (Tyner et al., 2018). For this regression task, an input sample was a patient's gene expression profile plus a one-hot encoded vector indicating which drug was tested in that patient, while the label we tried to predict was drug response (measured by IC50 - the concentration of the drug required to kill half of the patient's tumor cells). To define the graph

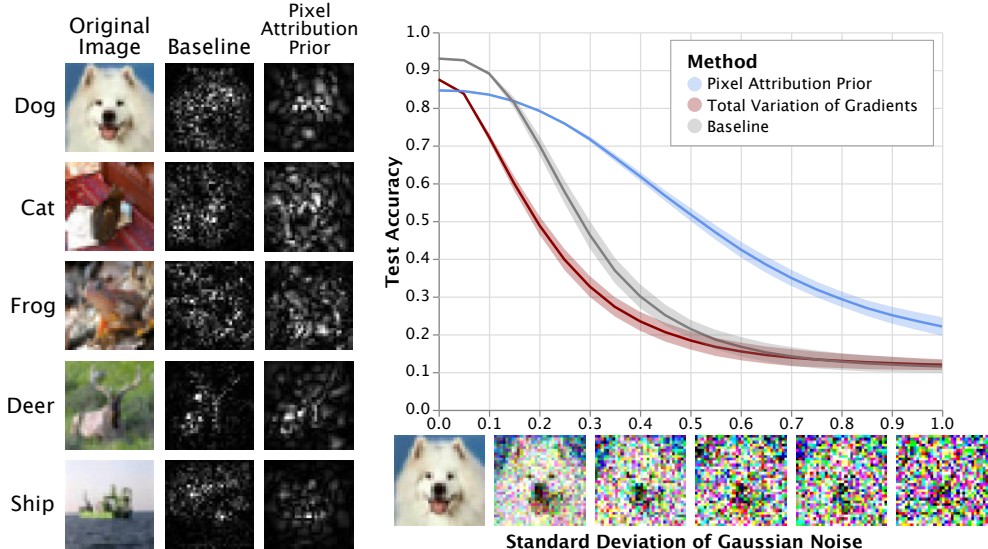

Figure 1: Left: Expected gradients attributions (from 100 samples) on CIFAR10 for both the baseline model and the model trained with an attribution prior, for five randomly selected images classified correctly by both the baseline and the regularized model. Training with an attribution prior generates visually smoother attribution maps in all cases. Notably, these smoothed attributions also appear more localized towards the object of interest. Right: Training with an attribution prior induces robustness to Gaussian noise, achieving more than double the accuracy of the baseline at high noise levels. This robustness isn't achievable by choosing gradients as the attribution function.

used by our prior we downloaded the tissue-specific gene interaction graph for the tissue most closely related to AML in the HumanBase database (Greene et al., 2015).

We find that a two-layer neural network trained with our graph attribution prior ($\Omega_{\text{graph}}$) significantly outperforms all other methods in terms of test set performance as measured by $R^2$ (Figure 2). Unsurprisingly, when we replace the biological graph from HumanBase with a randomized graph, we find that the test performance is no better than the performance of a neural network trained without *any* attribution prior. Extending the method proposed in Ross et al. (2017b) by applying our novel graph prior as a penalty on the model's *gradients*, rather than a penalty on the axiomatically correct expected gradient feature attribution, does not perform statistically significantly better than a baseline neural network. We also observe significantly improved test performance when using the prior graph information to regularize a linear LASSO model. Finally, we note that our graph attribution prior neural network significantly outperforms a recent method for utilizing graph information in deep neural networks, graph convolutional neural networks (Kipf and Welling, 2016).

To see if our model's attributions match biological intuition we conducted Gene Set Enrichment Analysis (a modified Kolmogorov–Smirnov test) to see if our top genes, as ranked by mean absolute feature attribution, were enriched for membership in any pathways (see the Appendix for more details, including the top pathways for each model) (Subramanian et al., 2005). We see that the neural network with the tissue-specific graph attribution prior captures significantly more biologically-relevant pathways (increased number of significant pathways after FDR correction) than a neural network without attribution priors (See Figure 2) (Benjamini and Hochberg, 1995). Furthermore, the pathways used by our model more closely match with biological expert knowledge – pathways included prognostically useful AML gene expression profiles, as well as important AML-related transcription factors (see Figure 2 and Appendix) (Liu et al., 2017; Valk et al., 2004).

### 4.3 A SPARSITY PRIOR IMPROVES PERFORMANCE WITH LIMITED TRAINING DATA

Here, we show that the $\Omega_{\text{sparse}}$ attribution prior can build sparser models that perform significantly better in settings with limited training data. We use a publicly available healthcare mortality prediction dataset of 13,000 patients (Miller, 1973), where the 36 features (119 after one-hot encoding) represent

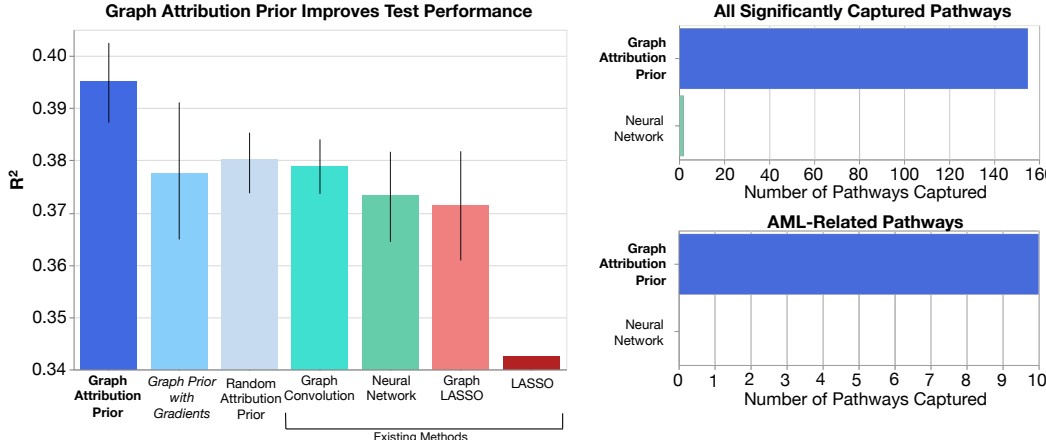

Figure 2: Left: A neural network trained with our graph attribution prior (bold) attains the best test performance, while a neural network trained with the same graph penalty on the gradients (italics, adapted from (Ross et al., 2017b)) does not perform significantly better than a standard neural network. Right: A neural network trained with our graph attribution prior has far more significantly captured biological pathways than a standard neural network, and also captures more AML-relevant pathways.

medical data such as a patient's age, vital signs, and laboratory measurements. The binary outcome is survival after 10 years. Sparse models in this setting may enable accurate models to be trained with very few labeled patient samples or reduce cost by accurately risk-stratifying patients using few lab tests. We subsample the training and validation sets to each contain only 100 patients, and run each experiment 100 times with a new random subsample to average out variance. We build 3-layer binary classifier neural networks regularized using L1, sparse group lasso (SGL) and sparse attribution prior penalties to predict patient survival, as well as an L1 penalty on gradients adapted for global sparsity from Ross et al. (2017b;a). The regularization strength was tuned from $10^{-10}$ to $10^3$ using the validation set for all methods, and the best model for each run was chosen using validation performance over 100 models trained with the chosen parameters (see Appendix).

The sparse attribution prior enables more accurate test predictions (Figure 3) and sparser models when little training data is available, with $p < 10^{-3}$ by Wilcoxon signed-rank test for all comparisons. We also plot the average cumulative importance of sorted features and find that the sparse attribution prior is much more effective at concentrating importance in the top few features (Figure 3). In particular, L1 penalizing the model's gradients as in Ross et al. (2017a) improves neither sparsity nor performance. A Gini gradient penalty slightly improves performance and sparsity but does not match the sparse attribution prior. Finally, we plot the average sparsity of the models (Gini coefficient) against their validation ROC-AUC across the full range of regularization strengths (Figure 3). The sparse attribution prior attains higher performance and sparsity than other models. Details and results for L2 penalties, dropout, and other attribution priors are in the Appendix.

## 5 RELATED WORK

There have been many previous attribution methods proposed for deep learning models (Lundberg and Lee, 2017; Binder et al., 2016; Shrikumar et al., 2017; Sundararajan et al., 2017). We chose to extend integrated gradients because it is easy to differentiate and comes with theoretical guarantees.

Training with gradient penalties has also been discussed by existing literature. Drucker and Le Cun (1992) introduced the idea of regularizing the magnitude of model gradients in order to improve generalization performance on digit classification. Since then, gradient regularization has been used extensively as an adversarial defense mechanism in order to minimize changes to network outputs over small perturbations of the input (Jakubovitz and Giryes, 2018; Yu et al., 2018; Roth et al., 2018). Ross and Doshi-Velez (2018) make a connection between gradient-based training for adversarial purposes and network interpretability. Ilyas et al. (2019) formally describe how the phenomena of

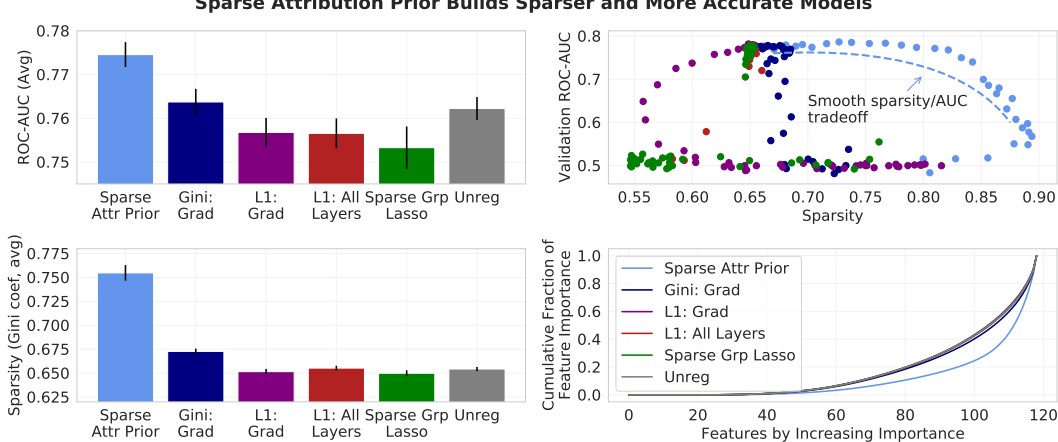

Figure 3: Left: A sparse attribution prior enables more accurate test predictions (top) and sparser models (bottom) across 100 small subsampled datasets (100 training and 100 validation samples each). Top right: Across the full range of tuned parameters, the sparse attribution prior achieves greater sparsity and a smooth sparsity-performance tradeoff. Bottom right: A sparse attribution prior concentrates a larger fraction of global feature importance in the top few features.

adversarial examples may arise due to features that are predictive yet non-intuitive, and stress the need to incorporate human intuition into the training process.

There is very little previous work on actually incorporating feature attribution methods into training. Sen et al. (2018) formally describe the problem of classifiers having unexpected behavior on inputs not seen in the training distribution, like those generated by asking whether a prediction would change if a particular feature value changed. They describe an active learning algorithm that updates a model based on points generated from a counter-factual distribution. Their work differs from ours in that they use feature attributions to generate counter-factual examples, but do not directly penalize the attributions themselves. Ross et al. (2017b) introduce the idea of training models to have correct explanations, not just good performance. Their method can be seen as a specific instance of our framework, in which the attribution function is gradients and the penalty function is minimizing the gradients of features known to be unimportant for each sample. Our work is more general in two ways. First, we instantiate three different penalty functions that encode human intuition without needing to know which features are unimportant in advance. Second, we propose a novel feature attribution method that can be regularized efficiently using a sampling procedure, and show that doing so provides better generalization performance than regularizing gradients with the same penalty.

## 6 DISCUSSION

The immense popularity of deep learning has driven its application in many domains with diverse, complicated prior knowledge. While it is in principle possible to hand-design network architectures to encode this knowledge, we propose a simpler approach. Using attribution priors, any knowledge that can be encoded as a differentiable function of feature attributions can be used to encourage a model to act in a particular way in a particular domain. We also introduce expected gradients, a feature attribution method that is theoretically justified and removes the choice of a single reference value that many existing feature attribution methods require. We further demonstrate that expected gradients naturally integrates with attribution priors via sampling during SGD. The combination allows us to improve model performance by encoding prior knowledge across several different domains. It leads to smoother and more interpretable image models, biological predictive models that incorporate graph-based prior knowledge, and sparser health care models that can perform better in data-scarce scenarios. Attribution priors provide a broadly applicable framework for encoding domain knowledge, and we believe they will be valuable across a wide array of domains in the future.

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

APPENDIX

## A   TRAINING WITH ATTRIBUTIONS

Normally, training with a penalty on any function of the gradients would require solving a differential equation. To avoid this, we adopt a double back-propagation scheme in which the gradients are first calculated with respect to the training loss, and alternately calculated with the loss with respect to the attributions (Yu et al., 2018; Drucker and Le Cun, 1992).

Our attribution method, expected gradients, requires background reference samples to be drawn from the training data. More specifically, for each input in a batch of inputs, we need $k$ additional inputs to calculate expected gradients for that input batch. As long as $k$ is smaller than the batch size, we can avoid any additional data reading by re-using the same batch of input data as a reference batch, as in Zhang et al. (2017). We accomplish this by shifting the batch of input $k$ times, such that each input in the batch uses $k$ other inputs from the batch as its reference values.

## B   CORRESPONDING PRIORS

In this section, we elaborate on the explicit form of the attribution priors used in the paper. In general, minimizing the error of a model corresponds to maximizing the likelihood of the data under a generative model consisting of the learned model plus parametric noise. For example, minimizing mean squared error in a regression task corresponds to maximizing the likelihood of the data under the learned model, assuming Gaussian-distributed errors.

$$\arg\min_{\theta} ||f_\theta(X) - y||_2^2 = \arg\max_{\theta} \exp(-||f_\theta(X) - y||_2^2) = \theta_{MLE}$$

where $\theta_{MLE}$ is the maximum-likelihood estimate of $\theta$ under the model $Y = f_\theta(X) + \mathcal{N}(0, \sigma)$.

An additive regularization term is equivalent to adding a multiplicative (independent) prior to yield a maximum a posteriori estimate:

$$\arg\min_{\theta} ||f_\theta(X) - y||_2^2 + \lambda||\theta||_2^2 = \arg\max_{\theta} \exp(-||f_\theta(X) - y||_2^2)\exp(-\lambda||\theta||_2^2) = \theta_{MAP}$$

Here adding an L2 penalty is equivalent to MAP for $Y = f_\theta(X) + \mathcal{N}(0, \sigma)$ with a $\mathcal{N}(0, \frac{1}{\lambda})$ prior. The natural next question is what attribution priors are being enforced by the penalties used in our experiments.

**Image prior**: Our image prior uses a total variation penalty, which has been well-studied.

$$\Omega_{\text{pixel}}(\Phi(\theta, X)) = \sum_{\ell} \sum_{i,j} |\phi^\ell_{i+1,j} - \phi^\ell_{i,j}| + |\phi^\ell_{i,j+1} - \phi^\ell_{i,j}|$$

It has been shown in Bardsley (2012) that this penalty is equivalent to placing 0-mean, iid, Laplace-distributed priors on the differences between adjacent pixel values. That is, $\phi^\ell_{i+1,j} - \phi^\ell_{i,j} \sim$ Laplace$(0, \lambda^{-1})$ and $\phi^\ell_{i,j+1} - \phi^\ell_{i,j} \sim$ Laplace$(0, \lambda^{-1})$. Bardsley (2012) does not call our penalty "total variation", but it is in fact the widely used anisotropic version of total variation, and is directly implemented in Tensorflow (Abadi et al., 2016; Lou et al., 2015; Shi and Chang, 2013).

**Graph prior**: The graph prior extends the image prior to arbitrary graphs:

$$\Omega_{\text{graph}}(\Phi(\theta, X)) = \bar{\phi}^T L_G \bar{\phi}$$

Just as the image penalty is equivalent to placing a Laplace prior on adjacent pixels in a regular graph, the graph penalty $\Omega_{\text{graph}}$ is equivalent to placing a Gaussian prior on adjacent features in an arbitrary graph with Laplacian $L_G$ (Bardsley, 2012).

**Sparsity prior**: Our sparsity prior uses the Gini coefficient as a penalty, which is written

$$\Omega_{\text{sparse}}(\Phi(\theta, X)) = -\frac{\sum_{i=1}^{p} \sum_{j=1}^{p} |\bar{\phi}_i - \bar{\phi}_j|}{n \sum_{i=1}^{p} \bar{\phi}_i} = -2G(\Phi)$$

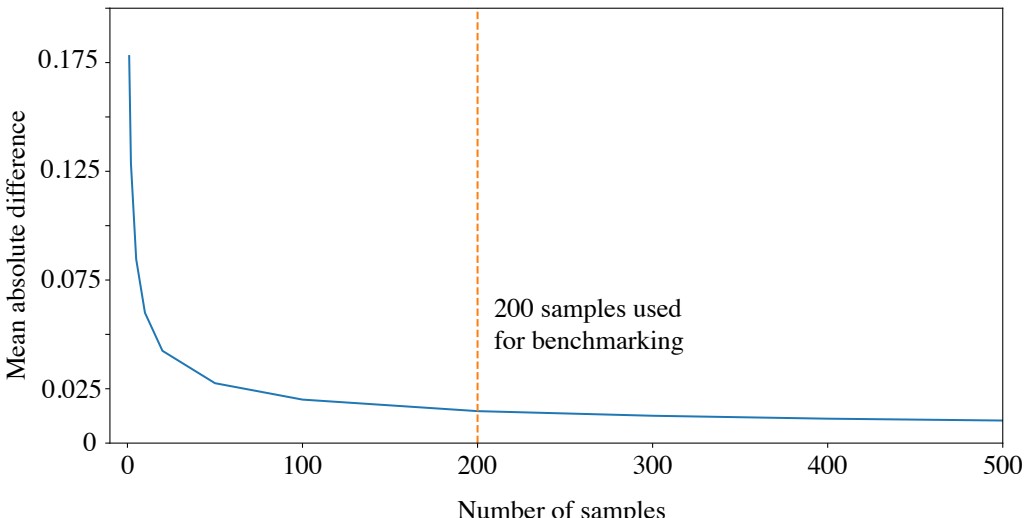

Figure 4: Feature attribution values attained using expected gradients converge as the number of background samples drawn is increased.

By taking exponentials of this function, we find that minimizing the sparsity regularizer is equivalent to maximizing likelihood under a prior proportional to the following:

$$\prod_{i=1}^{p}\prod_{j=1}^{p}\exp\left(\frac{1}{\sum_{i=1}^{p}\bar{\phi}_i}|\bar{\phi}_i - \bar{\phi}_j|\right)$$

To our knowledge, this prior does not directly correspond to a named distribution. However, we can note that its maximum value occurs when one $\bar{\phi}_i$ is 1 and all others are 0, as well as that its minimum occurs when all $\bar{\phi}_i$ are equal.

## C  BENCHMARKING EXPECTED GRADIENTS

### C.1  SAMPLING CONVERGENCE

Since expected gradients reformulates feature attribution as an *expected value* over two distributions (where background samples $x'$ are drawn from the data distribution and the linear interpolation parameter $\alpha$ is drawn from $U(0, 1)$), we wanted to ensure that we are drawing an adequate number of background samples for convergence of our attributions when benchmarking the performance of our attribution method. Since our benchmarking code was run on the Correlated Groups 60 synthetic dataset, as a baseline we explain all 1000 samples of this data sampling the full dataset (1000 samples) as background samples. To assess convergence to the attributions attained at this number of samples, we measure the mean absolute difference between the attribution matrices resulting from different numbers of background samples (see Figure 4). We empirically find that our attributions are well-converged by the time 100-200 background samples are drawn. Therefore, for the rest of our benchmarking experiments, we used 200 as the number of background samples. During training, even using the lowest possible setting of $k = 1$, we end up drawing far more than 200 background samples over the course of an epoch (order of magnitude in the tens of thousands, rather than hundreds).

Table 2: Benchmark on Independent Linear 60 dataset

| Attribution Method | KPM | KPR | KPI | KNM | KNR | KNI | KAM | KAR | KAI |
|---|---|---|---|---|---|---|---|---|---|
| Expected Gradients | **4.096** | **4.179** | **4.264** | **4.014** | **3.835** | **4.153** | **0.941** | **0.946** | **0.938** |
| Integrated Gradients | 4.055 | 4.112 | 4.176 | 3.949 | 3.753 | 4.070 | **0.941** | 0.945 | **0.938** |
| Gradients | 0.044 | 0.107 | 0.029 | 0.155 | -0.150 | 0.172 | 0.902 | 0.905 | 0.902 |
| Random | -0.152 | 0.102 | -0.152 | 0.111 | -0.126 | 0.060 | 0.470 | 0.482 | 0.438 |

| Attribution Method | RPM | RPR | RPI | RNM | RNR | RNI | RAM | RAR | RAI |
|---|---|---|---|---|---|---|---|---|---|
| Expected Gradients | **4.079** | **3.941** | **4.210** | **4.203** | **4.260** | **4.356** | **0.992** | **0.977** | **1.019** |
| Integrated Gradients | 4.013 | 3.854 | 4.113 | 4.157 | 4.186 | 4.259 | 0.973 | 0.966 | 0.995 |
| Gradients | 0.110 | -0.125 | 0.133 | 0.057 | 0.080 | 0.041 | 0.947 | 0.936 | 0.985 |
| Random | 0.012 | -0.124 | 0.059 | 0.035 | 0.101 | 0.070 | 0.504 | 0.521 | 0.527 |

## C.2 BENCHMARK EVALUATION METRICS

To compare the performance of expected gradients with other feature attribution methods, we used the benchmark metrics proposed in Lundberg et al. (2019). These metrics were selected as they capture a variety of recent approaches to quantitatively evaluating feature importance estimates. For example, the Keep Positive Mask metric (KPM) is used to test how well an attribution method can find the features that lead to the greatest increase in the model's output. This metric progressively removes features by masking with their mean value, in order from least positive impact on model output to most positive impact on model output, as ranked by the attribution method being evaluated. As more features are masked, the model's output is increased, creating a curve. The KPM metric measures the area under this curve (larger area corresponds to better attribution method). In addition to the KPM metric, 17 other similar metrics (e.g. Remove Absolute Resample, Keep Negative Impute, etc.) were used (see supplementary material of Lundberg et al. (2019) for more details on benchmark metrics). For all of these metrics, a larger number corresponds to a better attribution method. In addition to finding that Expected Gradients outperforms all other attribution methods on nearly all metrics tested for the dataset shown in Table 1 in the main text (the synthetic Correlated Groups 60 dataset proposed in Lundberg et al. (2019)), we also tested all 18 metrics on another dataset proposed in the same paper (Independent Linear 60) and find that Expected Gradients is chosen as the best method by all metrics in that case as well (see Table 2). The Independent Linear 60 dataset is comprised of 60 features, where each feature is a 0-mean, unit variance gaussian random variable plus gaussian noise, and the label to predict is a linear function of these features. The Correlated Groups 60 dataset is essentially the same, but now certain groups of 3 features have 0.99 correlation.

For attribution methods to compare, we considered expected gradients (as described in the main text), integrated gradients (as described in Sundararajan et al. (2017)), gradients, and random.

## D EXPECTED GRADIENTS ON IMAGENET

One unfortunate consequence of choosing an arbitrary baseline point for methods like integrated gradients is that the baseline point by definition is unimportant. That is, if a user chooses the constant black image as the baseline input, then purely black pixels will not be highlighted as important by integrated gradients. This is true for any constant baseline input. Since expected gradients integrates over a dataset as its baseline input, it avoids forcing a particular pixel value to be unimportant. To demonstrate this, we use the inception v4 network trained on the ImageNet 2012 challenge (Szegedy et al., 2017; Russakovsky et al., 2015). We restore pre-trained weights from the Tensorflow Slim library (Silberman and Guadarrama, 2016). In Figure 5, we plot attribution maps of both expected gradients and integrated gradients as well as raw gradients. Here, we use the constant black image as a baseline input for integrated gradients. For both attribution methods, we use 200 sample/interpolation points. The figure demonstrates that integrated gradients fails to highlight black pixels.

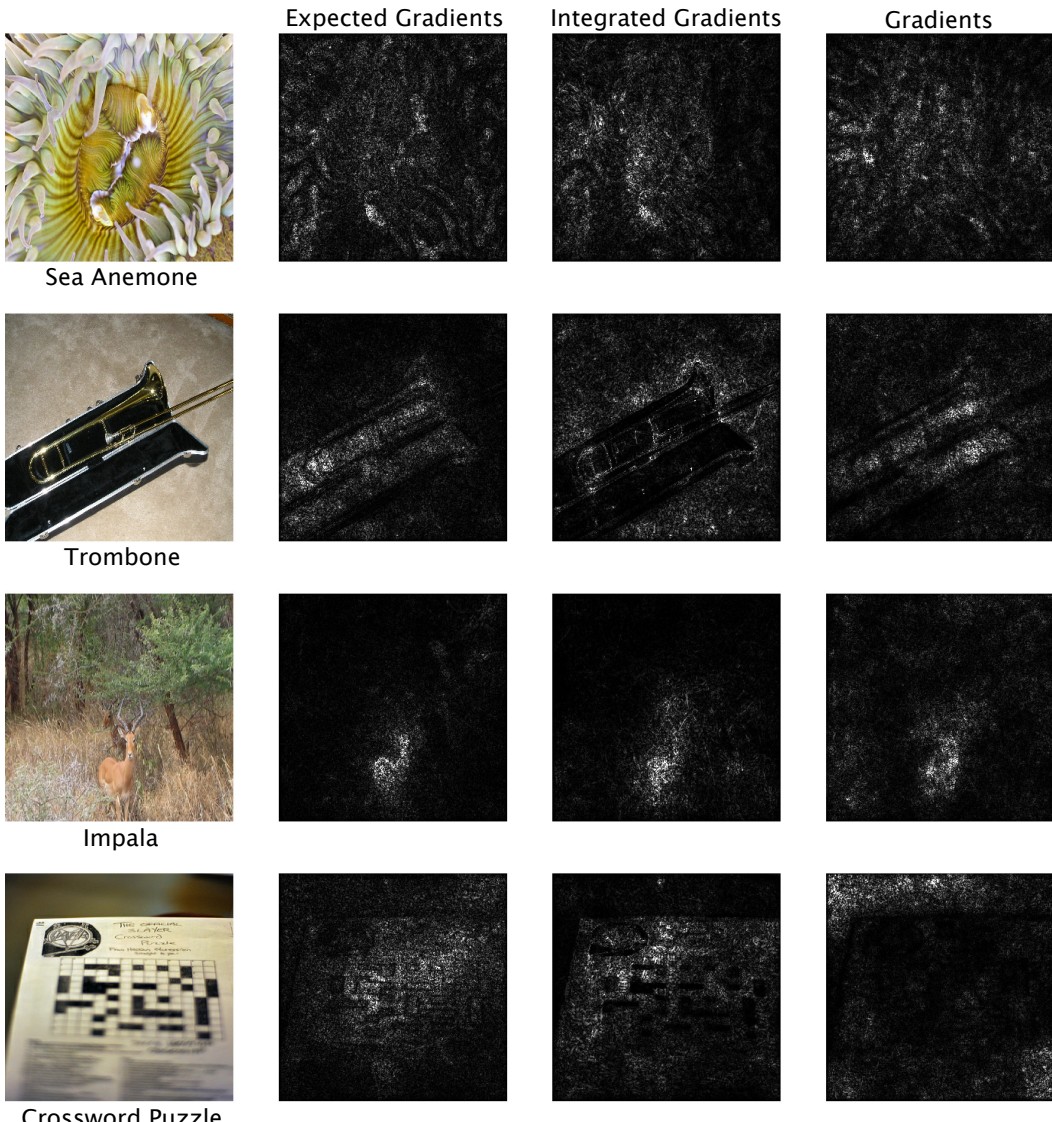

Figure 5: A comparison of attribution methods on ImageNet. Integrated gradients fails to highlight black pixels as important when black is used as a baseline input.

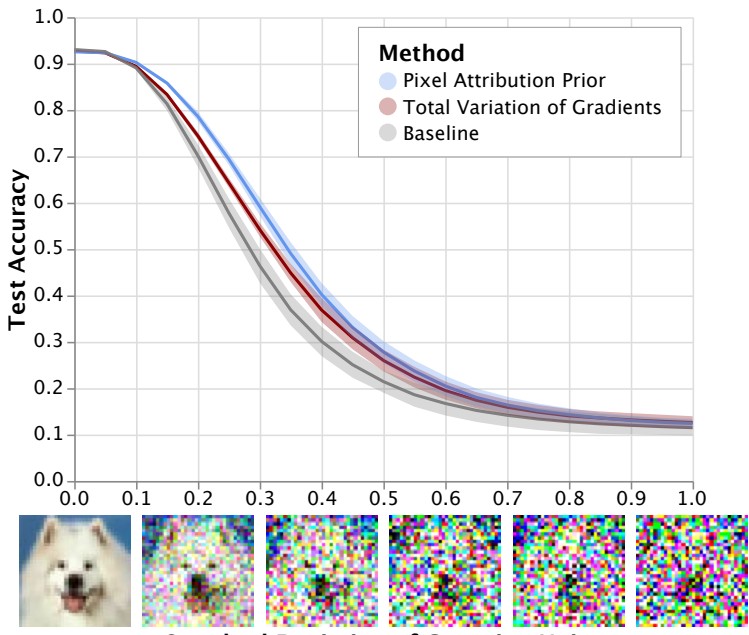

Figure 6: Robustness to noise on CIFAR-10 with a stricter $\lambda$ threshold. Here, there is little difference in test accuracy on the original test set between the baseline and the image attribution prior model ($0.930 \pm 0.002$ for the baseline vs. $0.925 \pm 0.002$ for the pixel attribution prior). Both the image attribution prior model and the gradient-based model afford small improvements in robustness compared to the baseline. As in the main text, results here are the mean and standard deviation across 5 random initializations.

# E  CIFAR-10 EXPERIMENTS

## E.1  EXPERIMENTAL SETUP

We train a VGG16 model from scratch modified for the CIFAR-10 dataset as in Liu and Deng (2015). We train using stochastic gradient descent with an initial learning rate of 0.1 and an exponential decay of 0.5 applied every 20 epochs. Additionally we use a momentum level of 0.9. For augmentation, we shift each image horizontally and vertically by a pixel shift uniformly drawn from the range [-3, 3], and randomly rotate each image by an angle uniformly drawn from the range [-15, 15]. We use a batch size of 128. Before training, we normalize the training dataset to have zero mean and unit variance, and standardize the test set with the mean and variance of the training set. We use $k = 1$ background reference sample for our attribution prior while training. When training with attributions over images, we first normalize the per-pixel attribution maps by dividing by the standard deviation before computing the total variation - otherwise, the total variation can be made arbitrarily small without changing model predictions by scaling down the pixel attributions close to 0

## E.2  CHOOSING LAMBDA

In the main text, we demonstrated the robustness of the image attribution prior model with $\lambda$ chosen as the value that minimized the total variation of attributions while keeping test accuracy within 10% of the baseline model. This corresponds to $\lambda = 0.001$ for both gradients and expected gradients if we search through 20 values logarithmically spaced in the range $[10^{-20}, 10^{-1}]$. If instead, we choose the $\lambda$ that minimizes total variation of attributions while keeping test accuracy equivalent to the baseline model (within 1%), we see that both the attribution prior and regularizing the gradients provides modest robustness to noise. This corresponds to $\lambda = 0.0001$ for both gradients and expected gradients. We show this result in Figure 6.

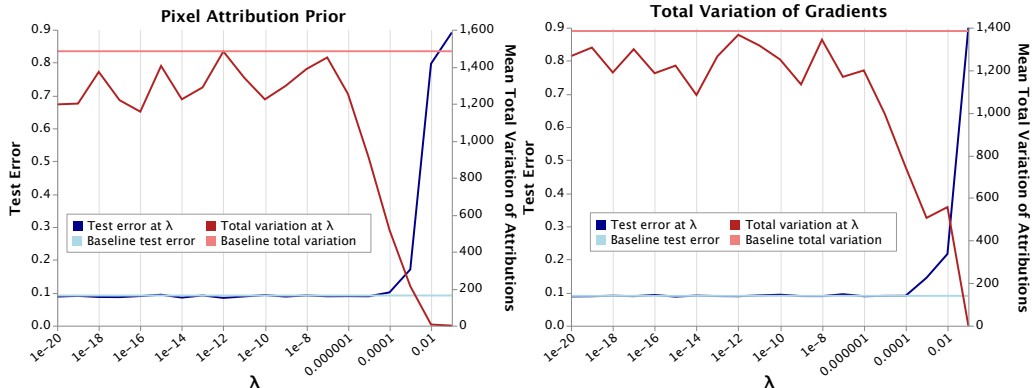

Figure 7: Plotting the trade-off between accuracy and minmizing total variation of expected gradients (left) or gradients (right). For both methods, there is a clear elbow point after which test accuracy degrades to no better than random. The total variation of attributions is judged based on the attribution being penalized: expected gradients for the left plot, gradients for the right plot.

For both the gradient-based model and the image attribution prior model, we also plot test accuracy and total variation of the attributions (gradients or expected gradients, respectively) in Figure 7. The $\lambda$ values we use correspond to the immediate two values before test accuracy on the original test set breaks down entirely for both the gradient and image attribution prior model.

## F  MNIST EXPERIMENTS

### F.1  EXPERIMENTAL SETUP

We repeat the same experiment on MNIST. We train a CNN with two convolutional layers and a single hidden layer. The convolutional layers have 5x5 filters, a stride length of 1, and 32 and 64 filters total, respectively. Each convolutional layer is followed by a max pooling layer of size 2 with stride length 2. The hidden layer has 1024 units, and a dropout rate of 0.5 during training (Srivastava et al., 2014). Dropout is turned off when calculating the gradients with respect to the attributions. We train with the ADAM optimizer with the default parameters ($\alpha = 0.001, \beta_1 = 0.9, \beta_2 = 0.999, \epsilon = 10^{-8}$) (Kingma and Ba, 2014). We train with an initial learning rate of 0.0001, with an exponential decay 0.95 for every epoch, for a total of 60 epochs. For all models, we train with a batch size of 50 images, and use $k = 1$ background reference sample per attribution while training.

### F.2  RESULTS

We choose $\lambda$ by sweeping over values in the range $[10^{-20}, 10^{-1}]$. We choose the $\lambda$ that minimizes the total variation of attributions such that the test error is within $1\%$ of the test error of the baseline model, which corresponds to $\lambda = 0.01$ for both the gradient model and the pixel attribution prior model. In Figure 8, we plot the robustness of the baseline, the model trained with an attribution prior, and the model trained by penalizing the total variation of gradients. We find that on MNIST, penalizing the gradients does similarly to penalizing expected gradients. We also find that it is easier to achieve high test set accuracy and robustness simultaneously.

### F.3  ATTRIBUTION MAPS

In Figure 8 we plot the attribution maps of the baseline model compared to the model regularized with an image attribution prior. We find that the model trained with an image attribution prior more smoothly highlights the digit in the image.

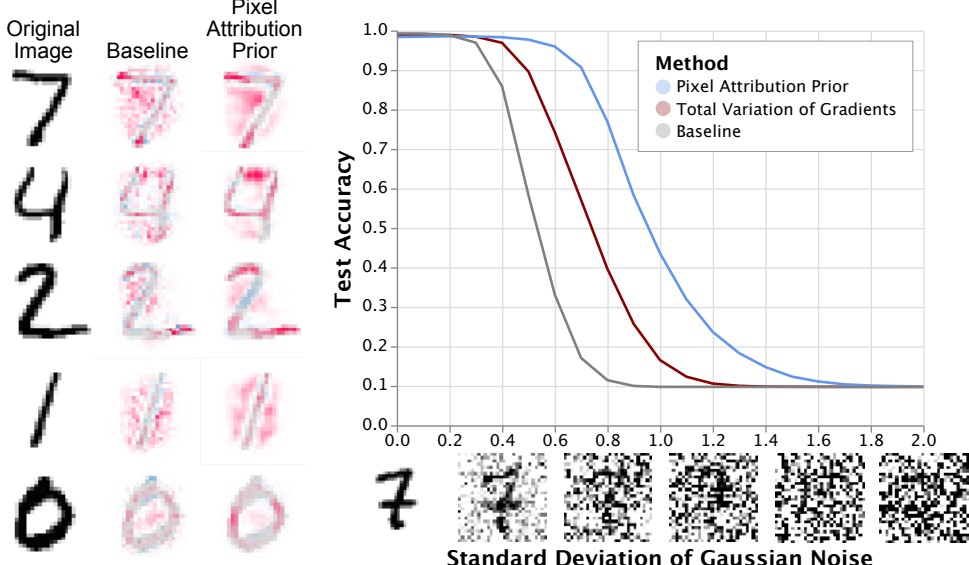

Figure 8: Left: Expected gradients attributions (with 100 samples) on MNIST for both the baseline model and the model trained with an attribution prior, for five randomly selected images classified correctly by both the baseline and the regularized model. Red pixels indicate pixels positively influencing the prediction, while blue pixels negatively influence the prediction. Training with an attribution prior generates visually smoother attribution maps and tend to better highlight relevant parts of the image. Right: Training with an attribution prior induces robustness to noise, more so than an equivalent model trained by minimizing the total variation of gradients or an equivalent baseline. The baseline model achieves an accuracy of 0.9925, compared 0.9836 for the pixel attribution prior and 0.9888 for the gradient model.

## G    IMAGENET EXPERIMENTS

In this section, we detail experiments performed on applying $\Omega_{\text{pixel}}$ to classifiers trained on the ImageNet 2012 challenge (Russakovsky et al., 2015). We omit this section from the main text since, for computational reasons, the hyperparameters chosen in this section may not necessarily be optimal.

### G.1    EXPERIMENTAL SETUP

We use the VGG16 architecture introduced by Simonyan and Zisserman (2014). For computational reasons, we do not train a model from scratch - instead, we fine-tune using pre-trained weights from the Tensorflow Slim package (Silberman and Guadarrama, 2016). We fine-tune on the ImageNet 2012 training set using the original cross entropy loss function in addition to $\Omega_{\text{pixel}}$ using asynchronous gradient updates with a batch size of 16 split across 4 Nvidia 1080 Ti GPUs. During fine-tuning, we use the same training procedure outlined by Silberman and Guadarrama (2016). This includes randomly cropping training images to $224 \times 224$ pixels, randomly flipping images horizontally, and normalizing each image to the same range. To optimize, we use gradient descent with a learning rate of 0.00001 and a momentum of 0.9. We use a weight decay of 0.0005, and set $\lambda = 0.00001$ for the first epoch of fine-tuning, and $\lambda = 0.00002$ for the second epoch of fine-tuning. As with the MNIST experiments, we normalize the feature attributions before taking total variation.

### G.2    RESULTS

We plot the attribution maps on images from the validation set using expected gradients for the original VGG16 weights (Baseline), as well as fine-tuned for 320,292 steps (Image Attribution Prior 1 Epoch) and fine-tuned for 382,951 steps, in which the last 60,000 steps were with twice the $\lambda$ penalty (Image Attribution Prior 1.25 Epochs). Figure 9 demonstrates that fine-tuning using our penalty results in sharper and more interpretable image maps than the baseline network. In addition, we also

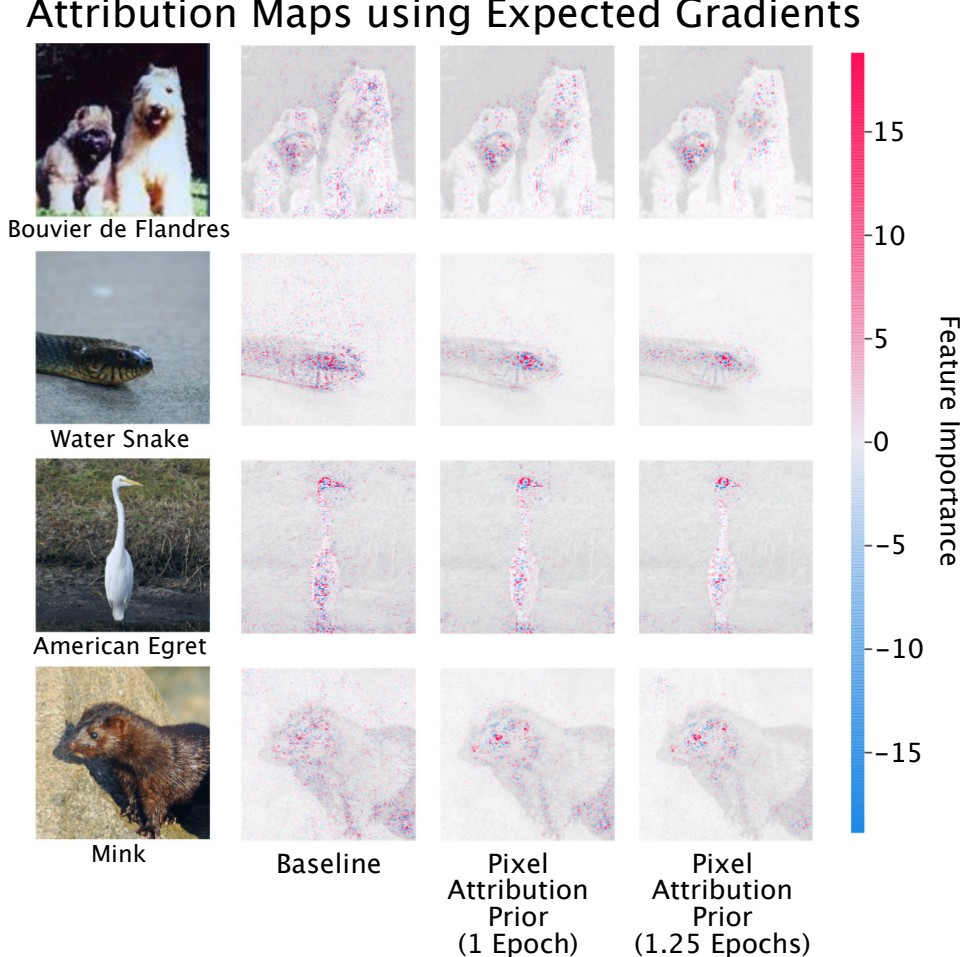

Figure 9: Attribution maps generated by Expected Gradients on the VGG16 architecture before and after fine-tuning using an attribution prior.

Table 3: Performance of the VGG16 architecture on the ImageNet 2012 validation dataset before and after fine-tuning.

| Model | Top 1 Accuracy | Top 5 Accuracy |
|---|---|---|
| Baseline | 0.709 | 0.898 |
| Image Attribution Prior 1 Epoch | 0.699 | 0.886 |
| Image Attribution Prior 1.25 Epochs | 0.674 | 0.876 |

plot the attribution maps generated by two other methods: integrated gradients (Figure 10) and raw gradients (Figure 11). Networks regularized with our attribution prior show more clear attribution maps using any of the above methods, which implies that the network is actually viewing pixels more smoothly, independent of the attribution method chosen.

We note that in practice, we observe similar trade-offs between test accuracy and interpretability/robustness mentioned in Ilyas et al. (2019). We show the validation performance of the VGG16 network before and after fine-tuning in Table 3 and observe that the validation accuracy does decrease. However, due to the computational cost even of fine-tuning on ImageNet, we did not perform a hyperparameter search for the optimal learning rate or $\lambda$ penalty. We anticipate that with more time and computational resources, we could achieve a better trade-off between interpretable attribution maps and test accuracy.

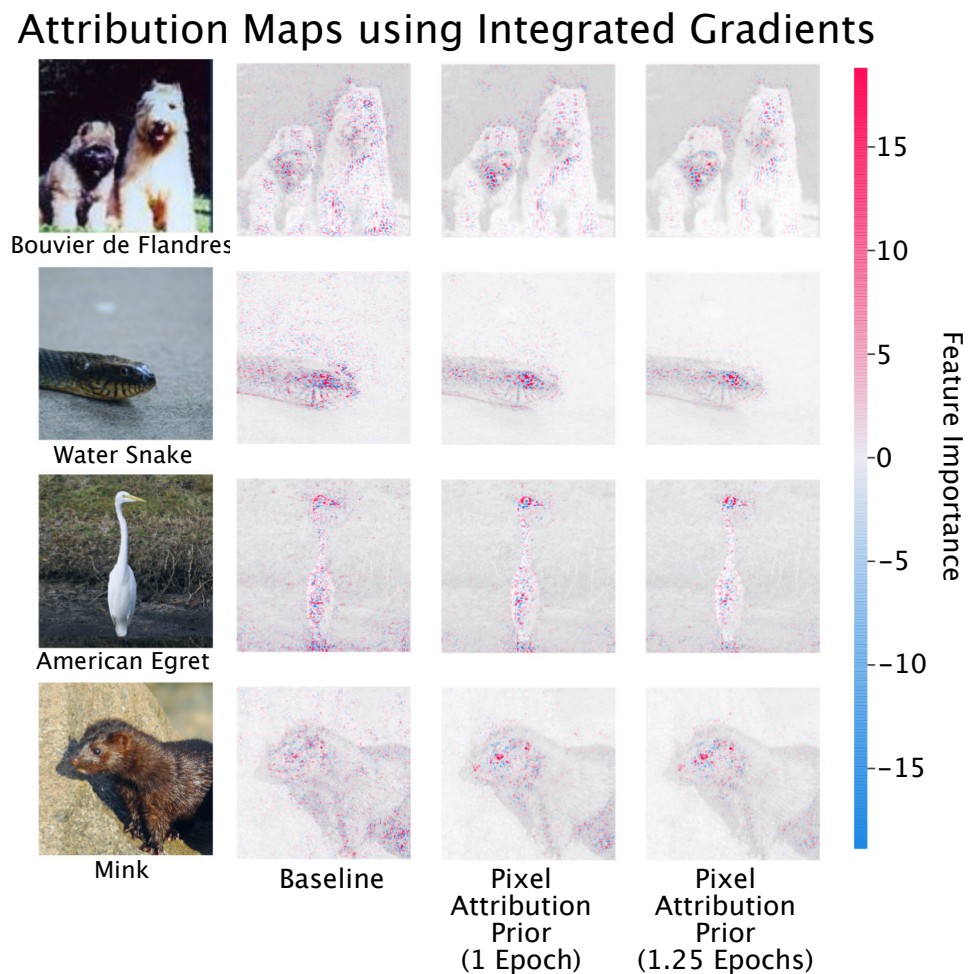

Figure 10: Attribution maps generated by Integrated Gradients on the VGG16 architecture before and after fine-tuning using an attribution prior.

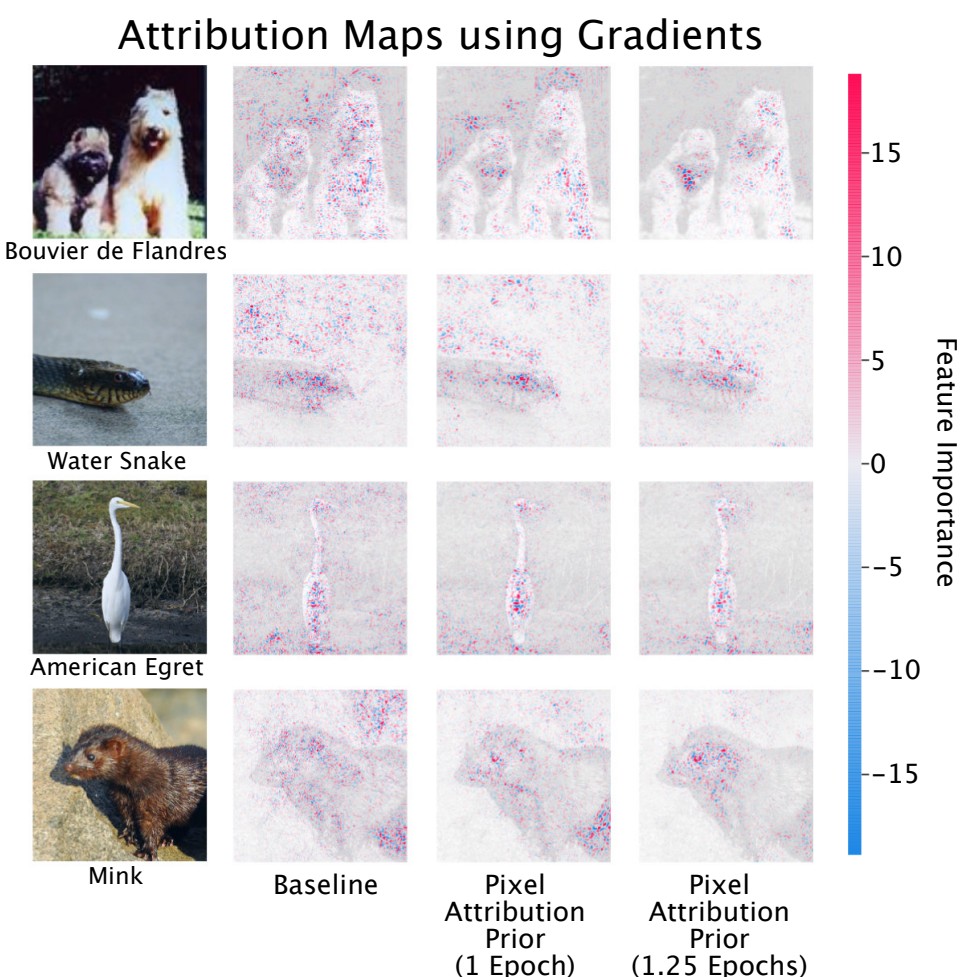

Figure 11: Attribution maps generated by raw gradients on the VGG16 architecture before and after fine-tuning using an attribution prior.

## H    Biological experiments

### H.1    RNA-seq preprocessing

To ensure a quality signal for prediction while removing noise and batch effects, it is necessary to carefully preprocess RNA-seq gene expression data. For the biological data experiments, RNA-seq were preprocessed as follows:

1. First, raw transcript counts were converted to fragments per kilobase of exon model per million mapped reads (FPKM). FPKM is more reflective of the molar amount of a transcript in the original sample than raw counts, as it normalizes the counts for different RNA lengths and for the total number of reads (Mortazavi et al., 2008). FPKM is calculated as follows:

$$FPKM = \frac{X_i \times 10^9}{N l_i} \tag{1}$$

   Where $X_i$ is the raw counts for a transcript, $l_i$ is the effective length of the transcript, and $N$ is the total number of counts.

2. Next, we removed non-protein-coding transcripts from the dataset.

3. We removed transcripts that were not meaningfully observed in our dataset by dropping any transcript where $> 70\%$ measurements across all samples were equal to 0.

4. We $\log_2$ transformed the data

5. We standardized each transcript across all samples, such that the mean for the transcript was equal to zero and the variance of the transcript was equal to one:

$$X_i^{'} = \frac{X_i - \mu_i}{\sigma_i} \tag{2}$$

   where $X_i$ is the expression for a transcript, $\mu_i$ is the mean expression of that transcript, and $\sigma_i$ is the standard deviation of that transcript across all samples.

6. Finally, we corrected for batch effects in the measurements using the ComBat tool available in the sva R package (Leek and Storey, 2007).

### H.2    Train / Validation / Test Set Allocation

To increase the number of samples in our dataset, we opted to use the identity of the drug being tested as a feature, rather than one of a number of possible output tasks in a multi-task prediction. This follows from prior literature on training neural networks to predict drug response (Preuer et al., 2018). This gave us 30,816 samples (covering 218 patients and 145 anti-cancer drugs). Defining a sample as a drug and a patient, however, meant we had to choose carefully how to stratify samples into our train, validation, and test sets. While it is perfectly legitimate in general to randomly stratify samples into these sets, we wanted to specifically focus on how well our model could learn trends from gene expression data that would generalize to novel patients. Therefore, we stratified samples at a patient-level rather than at the level of individual samples (e.g. no samples from any patient in the test set ever appeared in the training set). We split 20% of the total patients into a test set (6,155 samples), and then split 20% of the training data into a validation set for hyperparameter selection (4,709 samples).

### H.3    Model class implementations and hyperparameters tested

**LASSO:** We used the scikit-learn implementation of the LASSO (Tibshirani, 1996; Pedregosa et al., 2011). We tested a range of $\alpha$ parameters ranging from $10^{-9}$ to 1, and found that the optimal value for $\alpha$ was $10^{-2}$ by mean squared error on the validation set.

**Graph LASSO:** For our Graph LASSO we used the Adam optimizer in TensorFlow (Abadi et al., 2016), with a learning rate of $10^{-5}$ to optimize the following loss function:

$$\mathcal{L}(w; X, y) = \|Xw - y\|_2^2 + \lambda' \|w\|_1 + \nu' w^T L_G w \tag{3}$$

Where $w \in \mathbb{R}^d$ is the weights vector of our linear model and $L_G$ is the graph laplacian of our HumanBase network (Greene et al., 2015). In particular, we downloaded the "Top Edges" version of the hematopoietic stem cell network, which is thresholded to only have non-zero values for pairwise interactions that have a posterior probability greater than $0.1$. We used the value of $\lambda'$ selected as optimal in the regular LASSO model ($10^{-2}$, corresponds to the $\alpha$ parameter in scikit-learn), and then tuned over a range of $\nu'$ values ranging from $10^{-3}$ to $100$. We found that a value of $10$ was optimal according to MSE on the validation set.

**Neural networks:** We tested a variety of hyperparameter settings and network architectures via validation set performance to choose our best neural networks. We tested the following feed-forward network architectures (where each element in a list denotes the size of a hidden layer): [512,256], [256,128], [256,256], and [1000,100]. We tested a range of L1 penalties on all of the weights of the network, from $10^{-7}$ to $10^{-2}$. All models attempted to optimize a least squares loss using the Adam optimizer, with learning rates again selected by hyperparameter tuning from $10^{-5}$ to $10^{-3}$. Finally, we implemented an early stopping parameter of 20 rounds to select the number of epochs of training (training is stopped after no improvement on validation error for 20 epochs, and number of epochs is chosen based on optimal validation set error). We found the optimal architecture (chosen by lowest validation set error) had two hidden layers of size 512 and 256, an L1 penalty on the weights of $10^{-3}$ and a learning rate of $10^{-5}$. We additionally found that 120 was the optimal number of training epochs.

**Attribution prior neural networks:** To apply our attribution prior to our neural networks, after tuning our networks to the optimal conditions described above, we added extra epochs of fine-tuning where we ran an alternating minimization of the following objectives:

$$\mathcal{L}(\theta; X, y) = \|f_\theta(X) - y\|_2^2 + \lambda\|\theta\|_1 \tag{4}$$

$$\mathcal{L}(\theta; X) = \Omega_{graph}(\Phi(\theta, X)) = \nu\bar{\phi}^T L_G \bar{\phi} \tag{5}$$

Following Ross et al. (2017b), we selected $\nu$ to be $100$ so that the $\Omega_{graph}$ term would be initially equal in magnitude to the least squares and L1 loss term. We found that 5 extra epochs of tuning were optimal by validation set error. We drew $k = 10$ background samples for our attributions. To test our attribution prior using gradients as the feature attribution method (rather than expected gradients), we followed the exact same procedure only we now replace $\bar{\phi}$ with the average magnitude of the gradients rather than the average magnitude of the expected gradients.

**Graph convolutional networks:** We followed the implementation of graph convolution described in Kipf and Welling (2016). The architectures searched were as follows: in every network we first had a single graph convolutional layer (we were limited to one graph convolution layer due to memory constraints on each Nvidia GTX 1080-Ti GPU that we used), followed by two fully connected layers of sizes (512,256), sizes (512,128), or sizes (256,128). We tuned over a wide range of hyperparameters, including L2 penalties on the weights ranging from $10^{-5}$ to $10^{-2}$, L1 penalties on the weights ranging from $10^{-5}$ to $10^{-2}$, learning rates of $10^{-5}$ to $10^{-3}$, and dropout rates ranging from $0.2$ to $0.8$. We found the optimal hyperparameters based on validation set error were two hidden layers of size 512 and size 256, an L2 penalty on the weights of $10^{-5}$, a learning rate of $10^{-5}$, and a dropout rate of $0.6$. We again used an early stopping parameter and found that 47 epochs was the optimal number.

## H.4 DETAILS ON EXPERIMENTAL RESULTS

Looking at the resultant $R^2$ for prediction, we see that using the graph prior improves predictive performance of a linear model compared to L1-regularization alone (Graph LASSO vs. LASSO). However, we are able to attain a similar degree of predictive performance simply by switching from a linear model to a neural network that does not use the prior graph information at all. Our best performing model was the neural network with graph attribution prior. We use a $t$-test to compare the $R^2$ attained from 10 independent retrainings of the neural network to the $R^2$ attained from 10 independent retrainings of the attribution prior model and find that predictive performance is significantly higher for the model with the graph attribution prior ($p = 0.004696$).

Since we added graph-regularization to our model by fine-tuning, we wanted to ensure that the improved performance did not simply come from the additional epochs of training *without* the

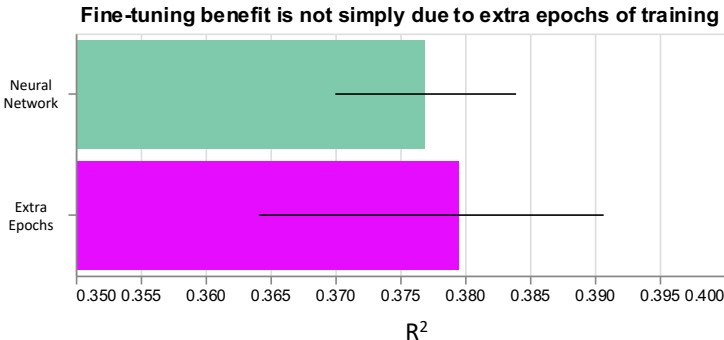

Figure 12: Fine tuning *without* graph prior penalty leads to no significant improvement in model performance.

attribution prior. We use a $t$-test to compare the $R^2$ attained from 10 independent retrainings of the regular neural network to the $R^2$ attained from 10 independent retrainings of the neural network with the same number of additional epochs that were optimal when adding the graph penalty (see Figure 12). We found no significant difference between the test error of these models ($p = 0.7565$).

To ensure that the increased performance in the attribution prior model was due to real biological information, we replaced the gene-interaction graph with a randomized graph (symmetric matrix with identical number of non-zero entries to the real graph, but entries placed in random positions). We then compared the $R^2$ attained from 10 independent retrainings of a neural network with no graph attribution prior to 10 independent retrainings of an neural network regularized with the random graph and found that test error was not significantly different between these two models ($p = 0.5039$). We also compared to graph convolutional neural networks, and found that our network with a graph attribution prior outperformed the graph convolutional neural network ($p = 0.0073$).

To ensure that the models were learning the attribution metric we tried to optimize for, we compared the explanation graph penalty ($\bar{\phi}^T L_G \bar{\phi}$) between the unregularized and regularized models, and found that the graph penalty was on average nearly two orders of magnitude lower in the regularized models (see Figure 14). We also examined the pathways that our top attributed genes were enriched for using Gene Set Enrichment Analysis and found that not only did our graph attribution prior model capture far more significant pathways, it also captured far more AML-relevant pathways (see Figure 13). We defined AML-relevant by a query for the term "AML," as well as queries for AML-relevant transcription factors.

## I  SPARSITY EXPERIMENTS

### I.1  DATA DESCRIPTION

Our data for the sparsity experiments used data from the NHANES I survey (Miller, 1973), and contained 36 variables (expanded to 119 features by one-hot encoding of categorical variables) gathered from 13,000 patients. The measurements include demographic information like age, sex, and BMI, as well as physiological measurements like blood, urine, and vital sign measurements. The prediction task is a binary classification of whether the patient was still alive (1) or not (0) 10 years after data were gathered.

### I.2  DATA PROCESSING

Data were mean-imputed and standardized so that each feature had 0 mean and unit variance. A fixed train/validation/test split of 7500/2500/3000 patients was used, with all hyperparameter tuning on the

**Most important genes for neural network *with attribution prior* come from biologically-relevant pathways**

| Pathway | FDR q-value |
|---|---|
| RNA Pol I Promoter Opening | < 10⁻²⁸⁰ |
| Amyloids | 0.002722 |
| Down-regulated in T Lymphocyte and NK Progenitor cells | 0.006435 |
| Down-regulated in normal aging | 0.007065 |
| TEL pathway | 0.007384 |
| B Cell Lymphoma Cluster 7 | 0.007601 |
| **AML Cluster 9** | 0.007604 |
| Response to MP470 up | 0.007853 |
| **Upregulated genes in cells immortalized by HOXA9 and MEIS1** | 0.008068 |
| **AML Cluster 12** | 0.008163 |

*… +145 more pathways*

**Most important genes for neural network *without attribution prior* are not significantly enriched for any AML-related pathways**

| Pathway | FDR q-value |
|---|---|
| RNA Pol I Promoter Opening | 0.001778 |
| Amyloids | 0.004001 |

*No additional pathways significant after FDR correction*

Figure 13: Top pathways for neural networks with and without attribution priors

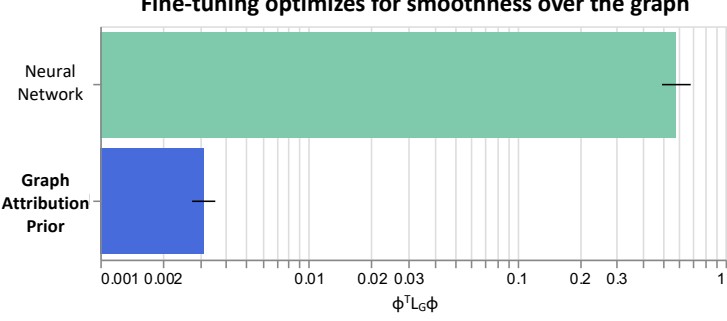

Figure 14: Fine tuning optimizes for the metric we care about: smoothness over the graph

validation set. For each of the 100 experimental replicates, 100 data points were sampled uniformly at random from the training and validation sets to yield a 100/100/3000 split.

## I.3    MODEL

We trained a range of neural networks to predict survival in the NHANES data. The architecture, nonlinearities, and training rounds were all held constant at values that performed well on an unregularized network, and the type and degree of regularization were varied. All models used ReLU activations and a 2-class softmax output; in addition, all models ran for 20 epochs with an SGD optimizer with learning rate 1.0 on the size-100 training data. The entire 100-sample training set fit in one batch. All 100 samples in the training set were used for expected gradients attributions during training and evaluation.

**Architecture:** We considered a range of architectures including single-hidden-layer 32-node, 128-node, and 512-node networks, two-layer [128,32] and [512,128]-node networks, and a three-layer [512,128,32]-node network; we fixed the [512,128,32] architecture for future experiments.

**Regularizers:** We tested a large array of regularizers. See section I.4 for details on how optimal regularization strength was found for each regularizer. *Italicized* entries were evaluated in the small-data experiments shown in the main text and I.6. For these penalties, the optimal regularization strength from validation-set tuning is listed. Non-italicized entries were evaluated on a sparsity/AUC plot using the full data (subsection I.7), but were not evaluated in the small-data experiments.

- *Sparse Attribution Prior* - $\Omega_{\text{sparse}}$ as defined in the main text. The best performing models for each replicate had an average regularization strength over the 100 runs of $\lambda = 1.60 \times 10^{-1}$.

- Mixed L1/Sparse Attribution Prior - Motivated by the observation that the Gini coefficient is normalized and only penalizes the *relative distribution* of global feature importances, we attempted adding an L1 penalty to ensure the attributions also remain small in an absolute sense. This did not result in improvements to performance or sparsity in full-data experiments (subsection I.7).

- *Sparse Group Lasso* - Rather than simply encouraging the weights of the first-layer matrix to be zero, the sparse group lasso also encourages entire columns of the matrix to shrink together by placing an L2 penalty on each column. As in Scardapane et al. (2017), we added a weighted sum of column-wise L2 norms to the L1 norms of each layer's matrix, without tuning the relative contribution of the two norms (equal weight on both terms). We also follow Scardapane et al. (2017) and penalize the absolute value of the *biases* of each layer as well. The average optimal regularization strength was $\lambda = 1.62 \times 10^{-2}$.

- *Sparse Group Lasso First Layer* - This penalty was similar to Scardapane et al. (2017), but instead of penalizing *all* weights and biases, only the first-layer weight matrix was penalized. This model outperformed the SGL implementation adapted from Scardapane et al. (2017), but did not outperform the sparse attribution prior, the Gini penalty on gradients, or the unregularized model. The average optimal regularization strength was $\lambda = 2.16 \times 10^{-3}$.

- L1 First-Layer - In order to facilitate sparsity, we placed an L1 penalty on the input layer of the network. No regularization was placed on subsequent layers.

- *L1 All Layers* - This penalty places an L1 penalty on all matrix multiplies in the network (not just the first layer). The average optimal regularization strength was $\lambda = 2.68 \times 10^{1}$.

- L1 Expected Gradients - This penalty penalizes the L1 norm of the vector of global feature attributions, $\bar{\phi}_i$ (analogous to how LASSO penalizes the weight vector in linear regression).

- L2 First-Layer - This penalty places an L2 penalty on the input layer of the network, with no regularization on subsequent layers.

- L2 All Layers - This penalty places an L2 penalty on all matrix multiplies in the network (not just the first layer).

- L2 Expected Gradients - This penalty penalizes the L2 norm of the vector of global feature attributions, $\bar{\phi}_i$ (analogous to how ridge regression penalizes the weight vector in linear models).

- Dropout - This penalty "drops out" a fraction $p$ of nodes during training, but uses all nodes at test time.

- *Baseline (Unregularized)* - Our baseline model used no regularization.

- *L1 Gradients* - To achieve the closest match to work by (Ross et al., 2017b;a), we placed a L1 penalty on the global gradients attribution vector of the network (mean across all samples of the absolute value of the gradient for each feature). This is similar to the "neural LASSO" of (Ross et al., 2017a), but with a goal of global sparsity (a model that uses few features overall) rather than local sparsity (a model that uses a small number of possibly different features for each sample). The average optimal regularization strength was $\lambda = 1.70 \times 10^{-2}$.

- *Gini Gradients* - An intermediate step between (Ross et al., 2017b;a) and our sparse attribution prior would use gradients as an attribution, but our Gini coefficient-based sparsity metric as a penalty. In this model we encouraged a large Gini coefficient of the mean absolute value of the gradients attributions of the model, averaged over all samples. The average optimal regularization strength was $\lambda = 1.33 \times 10^{-1}$.

The maintext figures, with small-data experiments repeated 100 times, compared the sparse attribution prior to methods previously used in literature on sparsity in deep networks – the L1 penalty on all layers, the sparse group lasso methods (Scardapane et al., 2017), and the L1 gradients penalty (Ross et al., 2017a). We also evaluated the Gini gradients penalty in these experiments. The other methods were not evaluated in the repeated small-data experiments shown in the maintext for space reasons, because there was less literature support, and because preliminary analysis (Figure 18) showed worse performance on sparsity with no benefit to accuracy.

## I.4    HYPERPARAMETER TUNING:

We selected the hyperparameters for our models based on the best validation performance over all parameters considered. There was one free parameter to tune for all methods other than the unregularized baseline (no tuning parameter) and the mixed L1/Sparse Attribution Prior model in our preliminary full-data experiments (two parameters - L1 and attribution penalty). We searched all L1, L2, SGL and attribution prior penalties with 131 points sampled on a log scale over $[10^{-10}, 10^3]$ (Figure 15). Some penalties, including the sparse attribution prior, mixed, gradient, and sparse group lasso penalties, produced NaN outputs for certain regularization settings. We retried several times when NaNs occurred, but if the problem persisted after multiple restarts, the parameter setting was skipped.

In preliminary experiments on the full data, we tuned the dropout probability with 130 points linearly spaced over $(0, 1]$. The mixed L1/Sparse Attribution Prior model was tuned in a 2D grid, with 11 L1 penalties sampled on a log scale over $[10^{-7}, 10^3]$ and 11 attribution prior penalties sampled on a log scale over $[10^{-10}, 10^0]$.

## I.5    MAINTEXT METHODS

**Performance and Sparsity Bar Plots:** The performance bar graph (Figure 3, top left) was generated by plotting mean test ROC-AUC of the best model of each type (chosen by validation ROC-AUC) averaged over each of the 100 subsampled datasets, with confidence intervals given by 2 times the standard error over the 100 replicates. The sparsity bar graph (Figure 3, bottom left) was constructed by the same process, but with Gini coefficients rather than ROC-AUCs.

**Feature Importance Distribution Plot:** The distribution of feature importances was plotted in the main text as a Lorenz curve (Figure 3, bottom right): for each model, the features were sorted by global attribution value $\bar{\phi}_i$, and the cumulative normalized value of the lowest $q$ features was plotted, from 0 at $q = 0$ to 1 at $q = p$. A lower area under the curve indicates more features have relatively small attribution values, indicating the model is sparser. Because 100 replicates were run on small subsampled datasets, the Lorenz curve for each model was plotted using the averaged mean absolute sorted feature importances, over all replicates. Thus, for a given model, the $q = 1$ point represented the mean absolute feature importance of the least important feature averaged over each replicate, $q = 2$ added the mean importance for the second least important feature averaged over each replicate, and so on.

**Performance vs Sparsity Plot:** Validation ROC-AUC and model sparsity were calculated for each of the 131 regularization strengths, and averaged over each of the 100 replicates. These were plotted

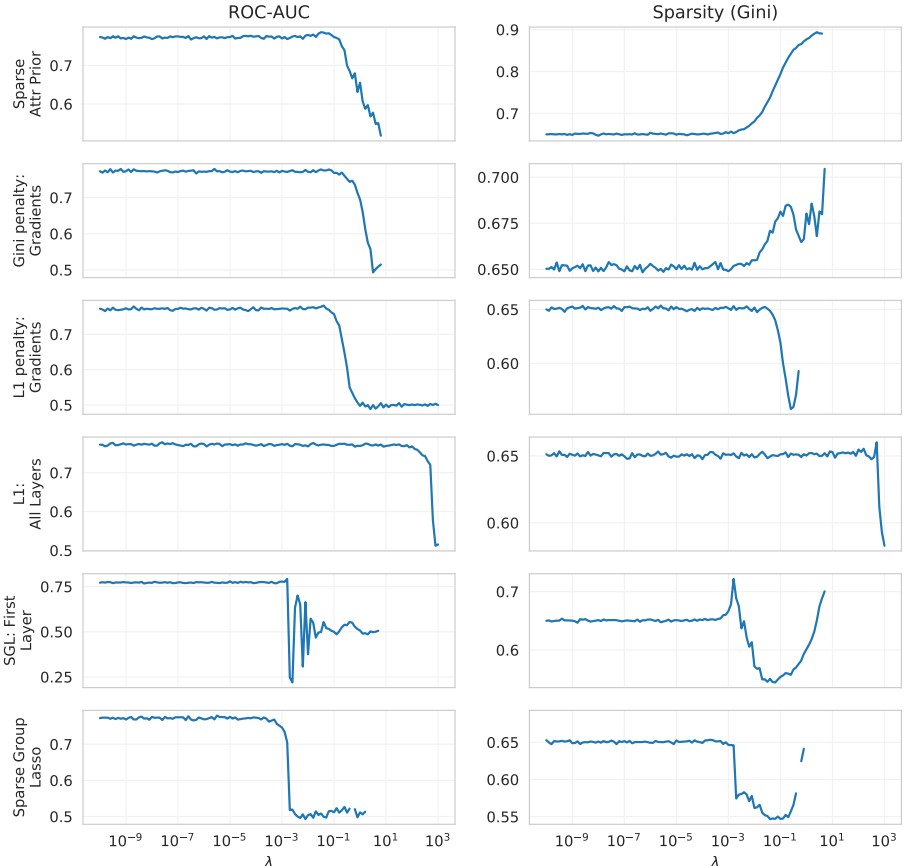

Figure 15: Validation performance and gini coefficient as a function of regularization strength for all models, averaged over 100 subsampled datasets. Blank areas indicate where some of the 100 models diverged for a given hyperparameter setting as described in subsection I.4.

on a scatterplot to show the possible range of model sparsities and ROC-AUC performances (Figure 3, top right), as well as the tradeoff between sparsity and performance. The sparse attribution prior was the only model capable of achieving a smooth tradeoff between sparsity and performance, as shown with the blue dashed line.

## I.6 ADDITIONAL RESULTS (MAINTEXT EXPERIMENTS)

**Statistical significance:** Statistical significance of the sparse attribution prior performance was assessed by comparing the ROC-AUCs of the best-performing sparse attribution prior models on each of the 100 subsampled datasets to those of the best-performing other models (L1 gradients, L1 weights, SGL, and unregularized). Significance was assessed by Wilcoxon signed-rank test, paired by subsampled dataset. The same process was used to calculate significance of model sparsity as measured by the Gini coefficient. The resulting $p$-values were:

|  | ROC | Gini |
|---|---|---|
| Gini penalty: Gradients | 5.49E-04 | 6.67E-13 |
| L1 penalty: Gradients | 2.55E-08 | 2.84E-15 |
| L1: All Layers | 2.95E-09 | 2.48E-14 |
| SGL: First Layer | 4.97E-05 | 2.48E-13 |
| Sparse Group Lasso | 4.40E-07 | 1.87E-15 |
| Unregularized | 1.76E-06 | 5.48E-15 |

**Additional SGL Penalty:** We show performance and sparsity for the penalties studied in the maintext plus first-layer SGL as bar plots, with confidence intervals from 100 experimental replicates (Figure 16 top two plots). The sparse attribution prior outperforms other methods by a wide margin. The Gini penalty on plain gradients performs slightly better than other methods, but not significantly. Thus it seems that the combination of both EG and Gini coefficient based penalties leads to better performance. The first-layer SGL slightly increases sparsity but does not outperform an unregularized model in ROC-AUC.

We also plot average performance on the validation set against average sparsity for the full range of searched parameters (Figure 16 bottom). Again, no method is able to compete with the sparse attribution prior in sparsity or performance, but the plain gradients Gini penalty also results in a small increase in sparsity, as do a small number of parameter settings for the first-layer SGL. There is a single point in the scatterplot for which first-layer SGL appears to outperform the sparse attribution prior in validation performance; however, this does not translate into superior test performance in the bar plots nor is there a smooth tradeoff curve between sparsity and AUC as with the sparse attribution prior.

**Feature Importance Summary:** We also show summaries of the mean absolute feature importance for the top 20 features in each model in Figure 17.

## I.7 ADDITIONAL RESULTS (ADDITIONAL PENALTIES)

We narrowed the range of possible penalties by studying the sparsity and performance achieved by additional penalties in preliminary experiments on the full dataset, without subsampling to study small-data performance. Performance (area under an ROC curve, AUC-ROC) was plotted as a function of sparsity (Gini coefficient) for all models. Figure 18 shows sparsity and validation performance for the same coarse initial parameter sweep as in the initial data, as well as sparsity and test performance for a fine sweep within the region of lowest cross-entropy for all models. The third image in the figure is a zoomed version to provide more detail on the best-performing models. The L1, SGL, and sparse attribution prior penalties were the best performing and the sparsest within these experiments.

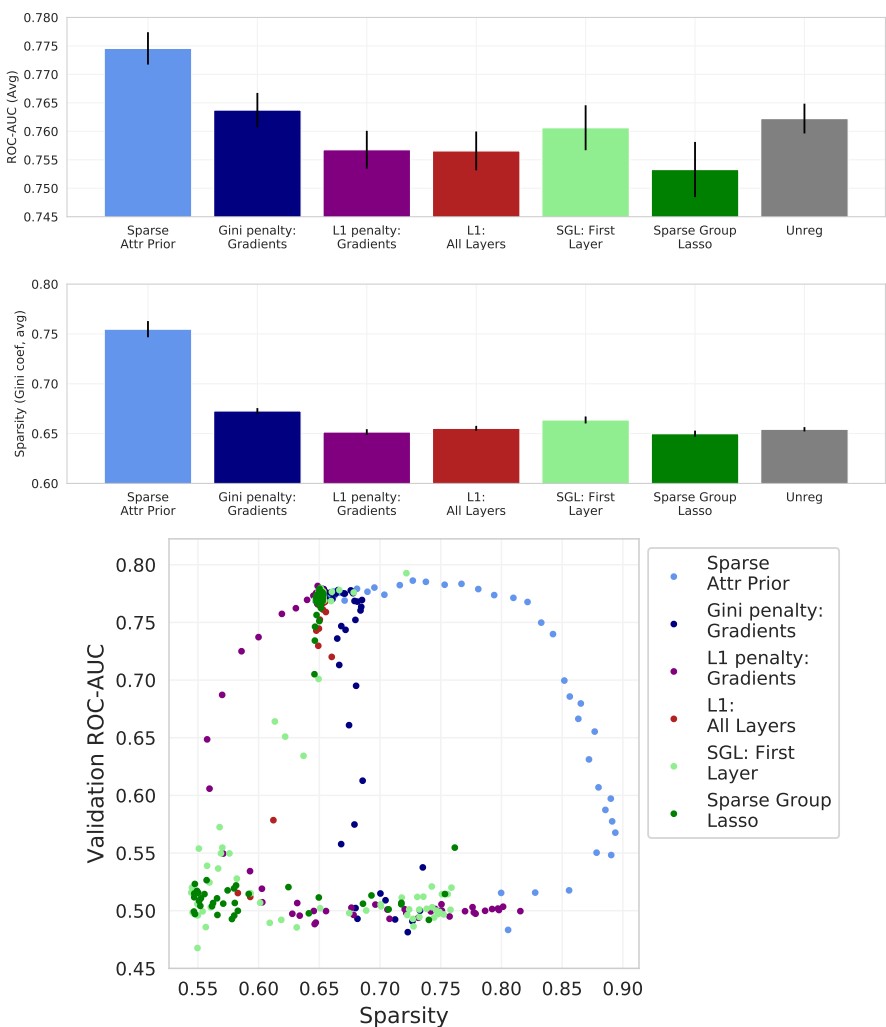

Figure 16: Additional results from the maintext experiments, with the addition of first-layer sparse group lasso. Top: The sparse attribution prior provides the best performance, and the Gini penalty on gradients provides the next best. First-layer SGL does not improve over unregularized models. Middle: The sparse attribution prior also builds the sparsest models, though the Gini gradients penalty also has slightly higher sparsity than the other models. First-layer SGL is slightly sparser than unregularized models. Bottom: Scatterplot of model sparsity and validation performance for all models in the maintext experiments, averaged across the 100 replicates. The sparse attribution prior achieves the highest performance for most parameters, though there is one parameter setting for which first-layer SGL outperforms it in validation loss (SGL does not end up winning in final test performance though, as seen in the bar plots). The only other model that often builds sparse models while maintaining performance is the Gini-based gradient penalty, though it is much less sparse.

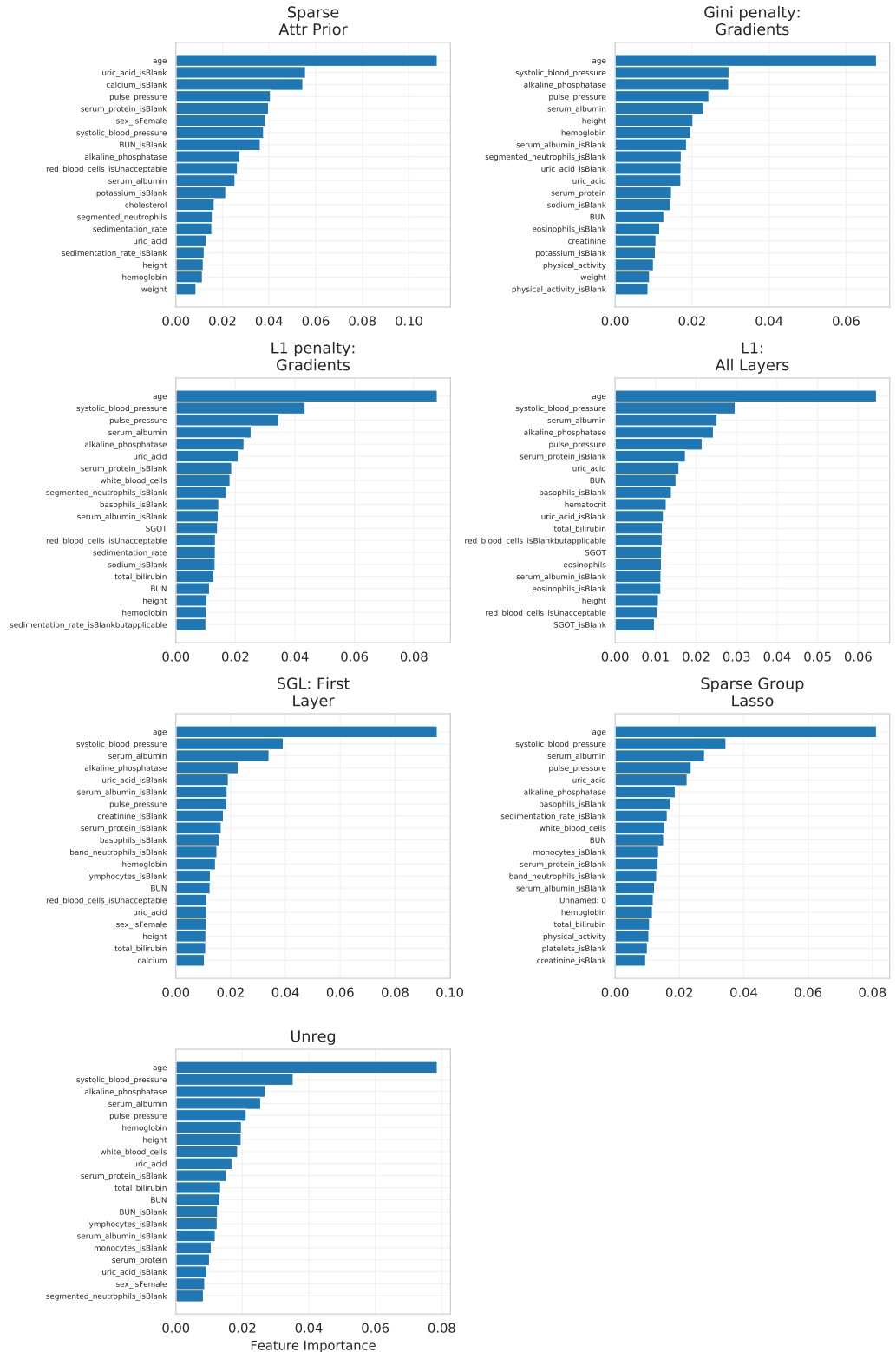

Figure 17: Summary of feature attributions for top 20 features from each model (best model from each class chosen as described in the main text), for a single randomly chosen replicate of the 100 small-data subsamples.

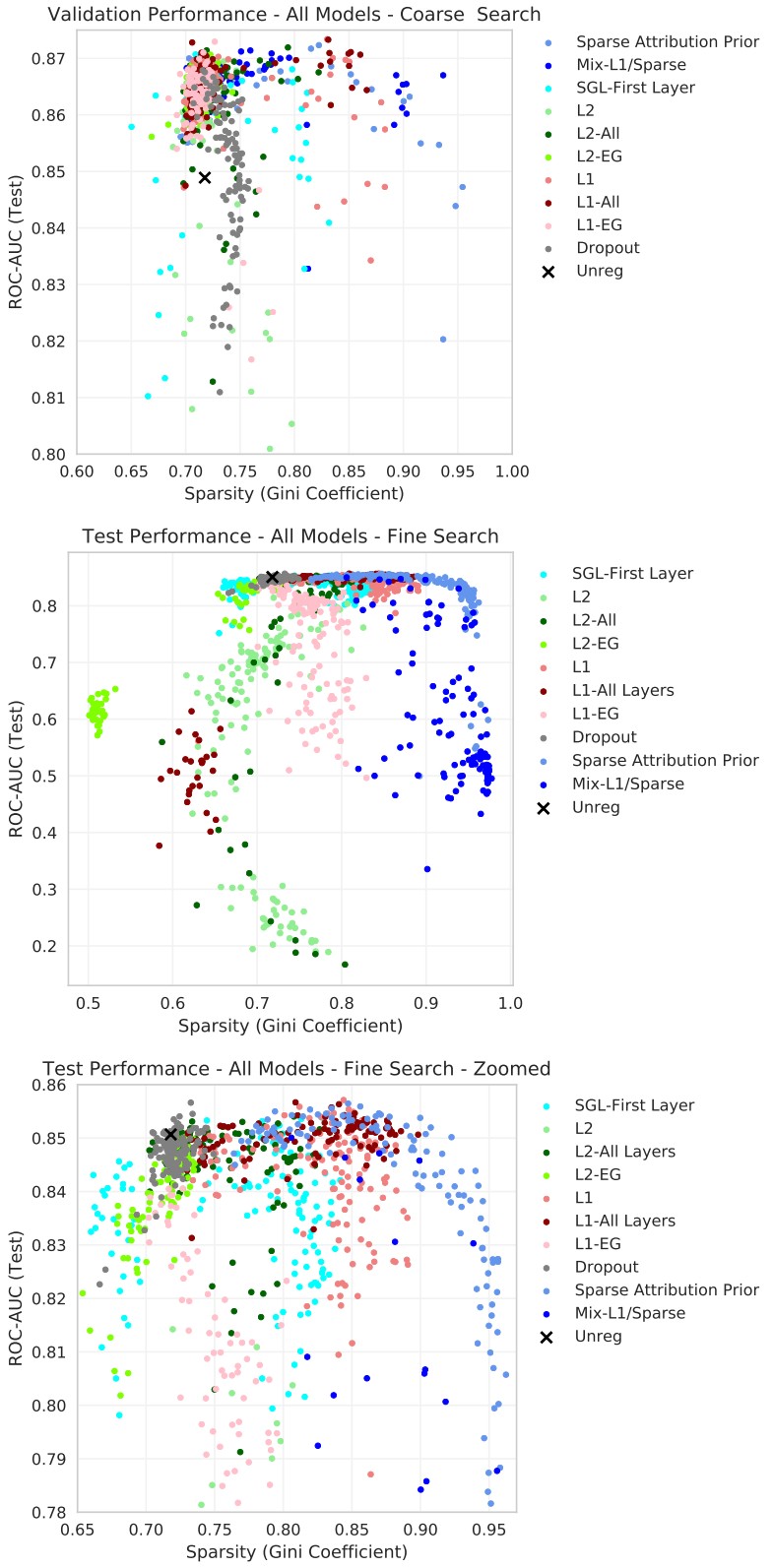

Figure 18: Sparsity vs performance plot for additional models on full NHANES dataset.

