# OpenReview forum: "Learning Explainable Models Using Attribution Priors"
_ICLR.cc/2020/Conference — Reject_

### Official Review · AnonReviewer2 · 2019-10-18
**Official Blind Review #2**

**Rating:** 8

**Review:**

The authors proposed the attribution priors framework to incorporate human domain knowledge as constraints when training deep neural networks. This is a general framework that the users can define different attribution priors for different tasks. For example, in this work, the authors proposed three reasonable priors for image input, graph data, and clinical medical data. For the image and the graph data, the prior is basically to have smoother attributions for nearby features; while for the clinical medical data, the authors used the Gini coefficient formula to encourage sparsity, which is of several practical benefits clinical practice. Moreover, the authors proposed the expected gradients algorithm which is a nice extension of the integrated gradients algorithm. The benefit of expected gradients is that it does not need a baseline input, which is usually arbitrary decided by the designer. The expected gradient method does indeed also performed better than the integrated gradient method in the benchmark (see Table 1.) The results in all three experiments are impressive. In the image domain, the model does generate models that paying more attention on the foreground objects, and is more tolerant to the Gaussian noise perturbation (though it does perform less well than a non-regularized baseline model in the no-noise test image, which is understandable.) More impressively, the model does outperform all other controls with a good margin in the anti-cancer drug prediction experiment, which is a nice demonstration of that domain knowledge could be incorporated in a neural network training to achieve better performance. Same to the healthcare mortality prediction data. The authors showed with a very limited amount of data, they can use sparsity prior constraints to get a model with good feature sparsity (Gini coef), and good performance (measured by ROC-AUC). Overall, I found the paper clearly written and the results are impressive. I am not super familiar with the field, and I am not sure how much progress is this paper compared to "Axiomatic Attribution for Deep Networks" (Sundararajan et. al. 2017),where integrated gradients is proposed. The experiments conducted in that paper seems to be similar to the ones that are done in this paper.

Minor point:
1. Even though the authors has shown in Table 1 benchmark that expected gradient is performing better than integrated gradient. Also, in Figure 5 showing that integrated gradient cannot highlight black pixels. It would be nice to see how integrated gradient method perform in the three experiments (image, drug data, mortality prediction), does the expected gradient method always outperform?
2. When the authors refer to Figure 2 and Figure 3 multiple times in the main text, they are referring to either left or right panel. Would be nicer to do for example "... as measured by R^2 (Figure 2 Left).

**Experience Assessment:**

I do not know much about this area.

**Review Assessment: Checking Correctness Of Derivations And Theory:**

I assessed the sensibility of the derivations and theory.

**Review Assessment: Checking Correctness Of Experiments:**

I carefully checked the experiments.

**Review Assessment: Thoroughness In Paper Reading:**

I read the paper at least twice and used my best judgement in assessing the paper.

---

> ### Comment · AnonReviewer1 · 2019-11-15
> **Are you sure there is novelty?**
>
> Given the existing paper https://arxiv.org/abs/1703.03717 what is significant about this paper? I don't see a significant contribution to justify an accept.

---

> > ### Author Response · Authors · 2019-11-15
> > **On the issue of novelty**
> >
> > Thank you for your concerns -- we hope that our comments in "Response to Official Blind Review #1" address some of these points.

---

> > ### Comment · AnonReviewer2 · 2019-11-15
> > **Thanks for the pointer**
> >
> > Thank you, AnonReviewer 1. I'm relative new to the field as claimed above. So, I may have not make the connection to that existing paper. I do find this paper is self-contained and well-written, which would is a good read to me. I totally respect your opinion that if there is nothing new compared to  https://arxiv.org/abs/1703.03717 then I would be happy to have my score ignored.

---

> ### Author Response · Authors · 2019-11-15
> **Official Blind Review #2**
>
> Thank you for pointing out some strengths of our paper. Sundarajan et al. (2017) certainly provide motivation for our work. We make two substantial contributions beyond the 2017 paper. First, we provide a better solution for the problem they present -- axiomatic attributions in deep networks -- by showing in Table 1 that Expected Gradients provides better explanations across a range of metrics. Second, we solve a new problem that Integrated Gradients did not discuss at all -- we show how to use the output of axiomatic attributions to highlight problems in a model’s behavior and use explicit priors to fix those problems.
>
> We do not compare against Integrated Gradients in our attribution prior experiments for computational reasons. Without re-formulating IG as an expectation, it is necessary to approximate a path integral at every training step, which would have required up to 50 additional back-propagation calls per training step, which was intractable for larger datasets. The reformulation as an expectation allows us to draw as little as one additional back-propagation call per training step and rely on regularizing the true attributions in expectation. This difficulty was in fact part of the motivation for developing and using Expected Gradients in our paper.
>
> Thanks for your comments on figure references; we will make the appropriate changes in the text.

---

### Official Review · AnonReviewer1 · 2019-10-23
**Official Blind Review #1**

**Rating:** 1

**Review:**

The paper presents expected gradients which is a method which looks at a difference from a baseline defined by the training data.

The structure of the paper is strange because it discusses attribution priors but then they are not used for the method. The paper should have a single focus.

Attribution priors as you formalize it in section 2 (which seems like the core contribution of the paper) was introduced in 2017 https://arxiv.org/abs/1703.03717 where they use a mask on a saliency map to regularize the representation learned.

In section 2.2. I think a few papers to have a look at are a survey article about graph based biasing http://www.nature.com/articles/s41698-017-0029-7 as well as methods for using graph convolutions with biases based on graphs: https://arxiv.org/abs/1711.05859 and https://arxiv.org/abs/1806.06975 . Some of these should serve as baselines. It is not clear which model is used in Figure 2. It is also not clear from the literature if these models are really working so I think these results should be presented in a more detail. As I understand it, real improvements in predicting clinical variables has not been shown to be reproducible so this would be a significant claim of this paper.

It is not clear if the paper is presenting "expected gradients" or existing attribution priors. Most of the experiments revolve around existing attribution prior methods. So with that the paper positions itself not as a survey but as a method paper but lacks evidence that the method expected gradients performs better.

I am also not clear on where the image attribution prior comes from for the image task. Where is this extra information? Is it just smoothing?


**Experience Assessment:**

I have published in this field for several years.

**Review Assessment: Checking Correctness Of Derivations And Theory:**

I assessed the sensibility of the derivations and theory.

**Review Assessment: Checking Correctness Of Experiments:**

I assessed the sensibility of the experiments.

**Review Assessment: Thoroughness In Paper Reading:**

I read the paper thoroughly.

---

> ### Author Response · Authors · 2019-11-15
> **Response to Official Blind Review #1**
>
> Thank you for the feedback. We acknowledge that Ross et al. (2017) proposed a specific instance of an attribution prior as discussed in Sections 1 and 2. However, we believe our contributions go significantly beyond the method proposed by Ross et al. (2017) and list our contributions below. We would appreciate any feedback on how we could better structure the manuscript to convey these points.
>
> 1) We introduce expected gradient attributions based on a novel expected-value based formulation that is specifically designed to allow for the efficient integration of attribution priors into the training process. We demonstrate both that expected gradients is a more reliable attribution method than existing methods (input gradients and integrated gradients) and that there exists a close connection between training with expected gradients and stochastic gradient descent that allows for efficient training of attribution priors (Section “Training with expected gradients”).
>
> 2) We demonstrate that our novel, expected gradients-based regularization method outperforms the method proposed in Ross et al. (2017) significantly in all three of our chosen applications. This advancement is likely due to our attribution method better capturing network behavior than simply the gradients themselves - it is well-understood that gradients do not reliably capture the behavior of a neural network (https://arxiv.org/abs/1611.02639, https://arxiv.org/abs/1704.02685).
>
> 3) We introduce 3 domain-specific attribution priors in three different domains that outperform unregularized baselines. The prior introduced in Ross et al. (2017) was limited to a binary penalty on features, meaning that for most applications, a user needed to know which features were important in advance. Our priors only require heuristic knowledge of the data and still improve performance.
>
> With respect to having two different focal points in our paper: we see expected gradients and attribution priors as closely related due to contributions 1) and 2). In addition to being a more reliable method than existing feature attribution methods, expected gradients allows for efficient training of an attribution prior because it is defined as an expectation. This efficiency allows us to improve over the method proposed in Ross et al. (2017). Without it, attribution priors would be limited to penalizing the raw gradients alone, which we demonstrate to be comparatively ineffective (See the red curve in Figure 1 Right, the second bar in Figure 2 Left, and the second bars in both of the left plots in Figure 3).
>
> With respect to our paper lacking evidence that the method expected gradients performs better than existing methods, we believe that Table 1 demonstrates that expected gradients is more effective at revealing model behavior than existing attribution methods, while Figures 1-3 demonstrate that training with expected gradients outperforms existing methods, including the one proposed in Ross et al. (2017) and several domain-specific baselines, such as the aforementioned graph convolutional networks.
>
> We will clarify the above points in the final version, and look forward to any feedback you may have on how best to present the contributions we describe above.

---

### Official Review · AnonReviewer3 · 2019-10-23
**Official Blind Review #3**

**Rating:** 3

**Review:**

Summary.
The paper improves the existing feature attribution method by adding regularizers to enforce (human) expectations about a model’s behavior. Three different datasets (i.e. image, gene expression, health-care) are chosen to evaluate the proposed model’s effectiveness, while different regularizers (i.e. image prior, graph prior, and sparsity prior) are explored for the respective task.

Strengths.
1. Incorporating human knowledge into the model has a growing interest in ML / CV communities.
2. Three datasets from different domains (i.e. image classification data, gene expression data, and health care data) are used to evaluate the effectiveness of the proposed approach. Data shows that the proposed approach shows better generalization performance (i.e. better performance in test dataset) than baselines.
3. The paper provides well-documented supplemental materials that contain details of the experimental setting and additional supporting figures.

Weaknesses.
1. Task-specific heuristic human prior
I agree (and personally like) the motivation that a method is needed to align a model’s behavior with human knowledge or intuition -- model’s behavior may be explained by feature attribution methods while making models accept human knowledge is challenging. However, such an ability is achieved by simply adding task-specific heuristic functions as a penalty or a regularizer. Also, the introduced human priors are similar to general regularization conventions, i.e. a penalty of smoothness over adjacent pixels is commonly used in the CV community. I am concerned that only a limited set of expert-invented human priors can be used in this approach.

Further, feature attribution methods aim to develop a richer notion of the contribution of a pixel to the output. However, the difficulty would be the lack of formal measures of how the network output is affected by spatially-extended features (rather than pixels). The explored priors (e.g. a total variation loss to make neighboring pixels have a similar impact on the final verdict) actually relieve this issue.

2. Incorporating humans into the modeling process?
A key motivation behind this work is “incorporating humans into the modeling process”. This would imply that (human-understandable) information needs first to be transferred from a model to humans. However, I am concerned about what information end-users are expected to obtain from the model. For example, Figure 1 (left) shows an attribution map that highlights multiple intermittent regions from which I cannot understand its behavior. Unless end-users cannot understand the model’s behavior, how can we expect humans can provide knowledge to model? A user study would be needed to support that the proposed method can really provide a way to incorporate humans into the modeling process.

Minor comments.
1. Plots in Figure 3 are not intuitively understandable.
2. There is no section Conclusion.
3. A template for the reference section looks different from other ICLR papers.

**Experience Assessment:**

I have read many papers in this area.

**Review Assessment: Checking Correctness Of Derivations And Theory:**

I assessed the sensibility of the derivations and theory.

**Review Assessment: Checking Correctness Of Experiments:**

I assessed the sensibility of the experiments.

**Review Assessment: Thoroughness In Paper Reading:**

I read the paper at least twice and used my best judgement in assessing the paper.

---

> ### Author Response · Authors · 2019-11-15
> **Response to Official Blind Review #3**
>
> Thank you for discussing several strengths of our paper. We believe these strengths represent a significant contribution toward addressing the issue of incorporating human knowledge into neural network models. Below, we discuss the weaknesses described in the review:
>
> 1) We first want to respond to the point that only a limited set of expert-invented human priors can be used in our approach. There are also only a limited number of expert-invented ways to regularize model parameters, yet parameter regularization (from parameter priors) is very important and widely studied in machine learning and statistics. In the same way we believe that regularizing feature attributions using expert-invented attribution priors promises to be a fundamentally new alternative to parameter regularization. Our paper significantly extends the pioneering work in this area by Ross et al (2017), and in doing so greatly expands the applicability of attribution priors.
>
> 2) In the context of incorporating human knowledge into machine learning models, we believe that one critical evaluation metric is how well our models do on prediction tasks. Our experiments show that our method is successful: it leads to improved performance in all three domains by using penalties derived from human intuition about the data. In terms of learning more intuitive models, we chose three task-specific metrics to optimize: smoothness in images, capturing related genes in gene expression data, and sparsity on clinical data. In all cases, we achieve our stated goal as evidenced by Figures 1-3. Whether or not these goals represent the most human-intuitive goals to optimize for in their respective domain would be valuable future work. In general, what constitutes an “interpretable” model is a challenging and open question. However, given the flexibility of our framework, we anticipate that it can be adapted to changing definitions of interpretability.

---

### Decision · Program_Chairs · 2019-12-19

**Decision:**

Reject

**Comment:**

This work claims two primary contributions: first a new saliency method "expected gradients" is proposed, and second the authors propose the idea of attribution priors to improve model performance by integrating domain knowledge during training. Reviewers agreed that the expected gradients method is interesting and novel, and experiments such as Table 1 are a good starting point to demonstrate the effectiveness of the new method. However, the claimed "novel framework, attribution priors" has large overlap with prior work [1]. One suggestion for improving the paper is to revise the introduction and experiments to support the claim "expected gradients improve model explainability and yield effective attribution priors" rather than claiming to introduce attribution priors as a new framework. One possibility for strengthening this claim is to revisit experiments in [1] and related follow-up work to demonstrate that expected gradients yield improvements over existing saliency methods. Additionally, current experiments in Table 1 only consider integrated gradients as a baseline saliency method, there are many others worth considering, see for example the suite of methods explored in [2].

Finally, I would add that the current section on distribution shift provides an overly narrow perspective on model robustness by only considering robustness to additive Gaussian noise. It is known that it is easy to improve robustness to Gaussian noise by biasing the model towards low frequency statistics in the data, however this typically results in degraded robustness to other kinds of noise types. See for example [3], where it was observed that adversarial training degrades model robustness to low frequency noise and the fog corruption. If the authors wish to pursue using attribution priors for improving robustness to distribution shift, it is important that they evaluate on a more varied suite of corruptions/noise types [4]. Additionally, one should compare against strong baselines in this area [5].

1. https://arxiv.org/abs/1703.03717
2. https://arxiv.org/abs/1810.03292
3. https://arxiv.org/abs/1906.08988
4. https://arxiv.org/abs/1807.01697
5. https://arxiv.org/abs/1811.12231